



# Glider-Based Observations of $CO_2$ in the Labrador Sea

Nicolai von Oppeln-Bronikowski[1], Brad deYoung[1], Dariia Atamanchuk[2], and Douglas Wallace[2]

[1]Department of Physics and Physical Oceanography, Memorial University, 283 Prince Phillip Drive, St. John's, NL, A1B3X7, Canada

[2]Department of Oceanography, Dalhousie University, 1355 Oxford Street, Halifax, NS, B3H4R2, Canada

**Correspondence:** Nicolai von Oppeln-Bronikowski (nbronikowski@mun.ca)

**Abstract.** Ocean gliders can provide high spatial and temporal resolution data and target specific ocean regions at a low cost compared to ship-based measurements. An important gap, however, given the need for carbon measurements, is the lack of capable sensors capable for glider-based $CO_2$ measurements. We need to develop robust methods to evaluate novel $CO_2$ sensors for gliders. Here we present results from testing the performance of a novel $CO_2$ optode sensor (Atamanchuk et al.,

2014), deployed on a Slocum glider, in the Labrador Sea and on the Newfoundland Shelf. We demonstrate our concept of validating data from this novel sensor during a long glider deployment using a secondary autonomous observing platform – the SeaCycler. Comparing data between different sensors and observing platforms can improve data quality and identify problems such as sensor drift. SeaCycler carried an extensively tested gas analyzer: the Pro-Oceanus's $CO_2$-Pro CV, as part of its instrument float. The $CO_2$-Pro CV has shown stable performance during lengthy observations e.g.(Jiang et al., 2014), but

has a slow response time for continuous profiling, and its power consumption is not affordable for glider operations. This $CO_2$ optode is an early prototype sensor that has not undergone rigorous testing on a glider, but is compact and uses little power. This paper summarizes the test results for this sensor on a Slocum glider. We capture the performance of the sensor, and for the Labrador Sea mission, comparing the glider data against the SeaCycler's measurements to compute an in-situ correction for the optode. We use the referenced data set to investigate trends in spatial and temporal variability captured by the glider data,

pointing to short time and distances scales as drivers of change in this region.

## 1 Introduction

The ocean plays a crucial role in absorbing the effects of changes to the Earth's atmospheric composition due to anthropogenic

activities. Roughly one-third of all human-made $CO_2$ ($C_{ant}$) released into the atmosphere since the beginning of the industrial revolution has been taken up by the ocean, a total of $155 \pm 31$ GtC as of 2010 (Khatiwala et al., 2013). For the decade 2009–2018 alone, the global ocean carbon sink absorbed $2.5 \pm 0.6$ GtC·yr$^{-1}$, against fossil fuel emissions of $9.5 \pm 0.5$ GtC·yr$^{-1}$ (Friedlingstein et al., 2019). Ocean carbon sinks are not equally distributed across the globe. Very intense carbon sinks and





regions of anthropogenic carbon storage are located in subpolar ocean regions (Volk and Hoffert, 1985; Sabine et al., 2004),
such as the Labrador Sea in the North Atlantic (DeGrandpre et al., 2006) and the Southern Ocean's Weddell Sea (van Heuven et al., 2014). Deep mixing in these regions is adding anthropogenic carbon to the deep ocean water mass transports, linking these high-latitude carbon pumps to the global ocean (Broecker, 1991; Fontela et al., 2016). Increased carbon storage in the ocean, has over the past decades, caused pH levels to drop in many places (Doney et al., 2009), at a rate of change that is faster than found in the geological record (Zeebe et al., 2016). Resulting ocean acidification (OA) has already severely impacted
marine habitats around the world, including such important ecosystems as the Great Barrier Reef (Cohen and Holcomb, 2009; Guinotte and Fabry, 2009).

Predicting shifts in future carbon uptake scenarios requires an in-depth understanding of the processes driving uptake and distribution of absorbed carbon, across all oceanic scales. We need to advance the global ocean carbon measurement system because existing observations are limited in coverage and quality (Borges et al., 2010; Okazaki et al., 2017). There have been
recent advances in autonomous sampling strategies to expand, improve and build-on existing global biogeochemical observing networks (Johnson et al., 2009). The existing Argo float program is being expanded to include biogeochemical (BGC-Argo) sensors measuring oxygen, nitrate, chlorophyll, turbidity, irradiance and pH. BGC-Argo is aimed at observing seasonal to decadal-scale variability, although currently only about 8% of Argo floats are equipped with biogeochemical sensors (Johnson et al., 2017; Li et al., 2019). Improvements in resolution and frequency of surface $CO_2$ measurements have also been made
with the development of stable ship-based in-situ measurement systems installed on container ships and tankers with regular routes across ocean basins. These results made possible the creation of a 1° global resolution (up to 1/4° coastal zones) Surface Ocean $CO_2$ Atlas (Bakker et al., 2016). However, these data do not provide researchers with the information at depth needed to understand the localized processes that drive and shape the strength of carbon sinks regions such as the Labrador Sea. Advances in gliders technology and sensors (Rudnick, 2016; Testor et al., 2019) can help address those gaps.

Advancing glider-based measurements of $CO_2$ requires addressing key issues such as stability, responsiveness, compactness and power-consumption. (Clarke et al., 2017a, b; Fritzsche et al., 2018). So far, most carbon glider observations are limited to testing and there remain concerns about data quality. The most mature and commonly used type of in-situ $CO_2$ probe is based on infrared (IR) detection, such as the CONTROS Hydro C™ or Pro Oceanus $CO_2$-Pro CV™ sensors. Unfortunately, commercial IR based detection systems are not yet small enough to easily fit on existing gliders or float designs. Long equilibration
times make profiling application of sensors extremely challenging, requiring detailed knowledge of response times and data processing (Fiedler et al., 2013; Atamanchuk et al., 2020). These sensors are also very power hungry compared to other sensors like optodes or CTD's, making battery-powered deployments challenging even for moored applications. Another approach to determining in-situ $CO_2$ is through pH measurements using established Total Alkalinity (TA) and Salinity (S) relationships (Takeshita et al., 2014). Saba et al. (2018) applied a novel ISFET pH (Johnson et al., 2016) sensor developed by MBARI with
help from Sea-Bird Scientific on a glider. These tests showed remarkable response time characteristics and stability over a period of several weeks or longer.

Another candidate for glider carbon observations is the Aanderaa $CO_2$ optode sensor (Atamanchuk et al., 2014). It is nearly identical in size and power consumption to the commonly used oxygen optode by the same company but lacks prior glider





testing. The optode detects the luminescent-quenching response from $CO_2$ sensitive membrane. In general, there are multiple
challenges to using photo-chemical sensors on profiling applications Bittig et al. (2014): (1) placement of the sensor on the
glider dictates boundary layer thickness and response time; (2) response time is non-linearly temperature-dependent and steep
temperature gradients induce additive error; and (3) the sensor is highly dependent on prior foil calibration and can suffer
from drift. In particular, the foil design has multiple temperature-dependent rate-limiting processes inside the foil to sense the
ambient change in pH, which is correlated to changes in $pCO_2$ (S. M. Borisov, *personal communications*). On the upside, the
$CO_2$ optode is an attractive candidate for gliders, due to its small size, ease of integration, and low power consumption, all
similar to the Aanderaa oxygen optode. Because of the need for increased spatial and temporal resolution of $CO_2$ observations
and the advantages gliders offer compared to other methods, assessing the $CO_2$ optode on a glider is an important step in
furthering community knowledge on the current state of mobile $CO_2$ system technology.

In 2016, as part of the Ventilations, Interactions and Transports Across the Labrador Sea (VITALS) project, we devised an
observing strategy to carry out novel in-situ observations to: (1) Reach the deep convection region with a glider to carry out
sampling with the novel foil-based $pCO_2$ sensor from Aanderaa with minimal ship resources for launch and recovery. (2) Use
measurements provided by an autonomous moored profiler - the SeaCycler (Send et al., 2013), carrying the larger payload $CO_2$-
Pro CV instrument for glider in-situ calibration points. This mission attempted to use a moored sensing platform as an in-situ
reference point for experimental sensors deployed on a glider to advance data quality and coherence of novel biogeochemical
measurements. This is an important step towards targeted oceanic carbon measurements as technology is playing a catch-up
game. We extended this mission in September 2018 on the Newfoundland Shelf, in Trinity Bay, to further test the concept,
flying the glider near a small fishing boat from which reference casts were taken using a similar Pro CV instrument. We utilize
these two real ocean deployments to improve sensor characterization and the quality of the collected data. In this paper, we
present the data and our analysis and discussion around three central questions:

– How suitable is the $CO_2$ optode for glider-based applications?

– How can multiple autonomous platforms be used to improve sensor data?

– How can combined data from moored and mobile platforms resolve scales of temporal and spatial variability?

Addressing these questions should improve and shape our current plans for carbon observing systems utilizing glider and other
platforms, especially as new sensors are being developed.

## 2 Data and Methods

### 2.1 Labrador Sea Deployment

In the fall 2016, a moored vertical profiler, the SeaCycler (Send et al., 2013; Atamanchuk et al., 2020) and a G2 Slocum
glider were deployed into the central Labrador Sea near the longtime German deep convection mooring K1 (Figure 1). The K1
mooring, located about 25 km west of former OWS BRAVO (Avsic et al., 2006), has been deployed biennially since 1994 to





monitor activity in the central deep convection patch in the Labrador Sea (Lavender et al., 2002; Koelling et al., 2017). The
objective of VITALS was to characterize the spatial and temporal structure of oxygen and $CO_2$ in the deep convection zone.
Other activity in conjunction with VITALS, included a hydrographic section AR7W maintained by the Bedford Institute of
Oceanography (BIO) and Argo floats released with several profiles captured near the SeaCycler site and the glider deployment
area. Many observing efforts came together, utilizing multiple complementary efforts across different scientific programs,
relying on both traditional and novel observational approaches.

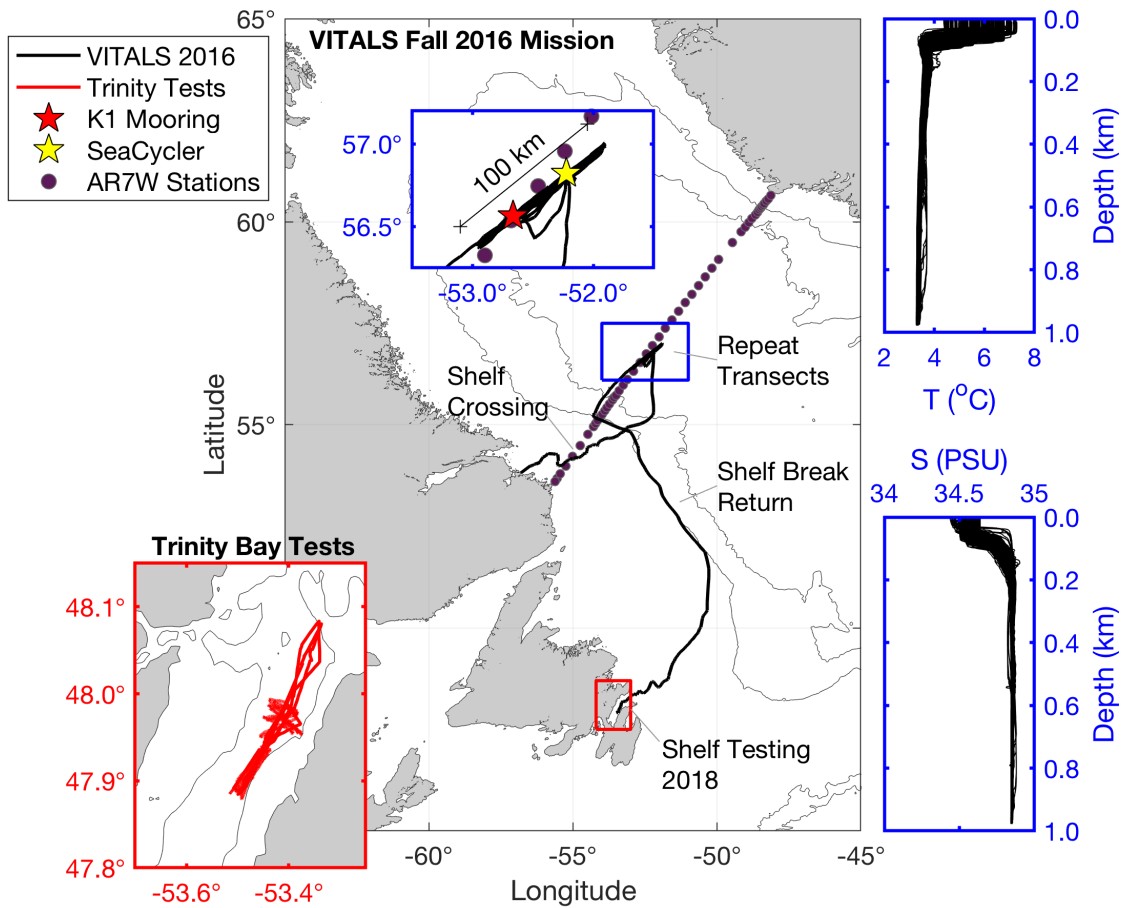

**Figure 1.** Map of data collection sites.

The SeaCycler was deployed near 52.22°W and 56.82°N, 30 km away from German deep convection mooring K1 (52.66°W
and 56.56°N) to improve the vertical and temporal characterization of $O_2$ and $CO_2$ cycling in this region. The SeaCycler
operation and deployment techniques are described in Send et al. (2013). It has an underwater winch assembly, parked at 160
m depth with an instrument float that can profile the top 150 m. A tethered communication allows for two-way telemetry
over Iridium Satellite. Below the winch assembly, a single-point mooring line with instruments continues to the ocean depth





of approximately 3500 m. For this deployment, the instrument float carried CTD, velocity and various gas sensors, including oxygen sensors (Sea-Bird 43, Sea-Bird 63, Aanderaa 4330) and pCO2 sensor $CO_2$ Prototype 4797) as well as membrane equilibrator-based infrared (IR) $CO_2$-Pro CV, based on non-dispersive infrared refraction (NDIR) technology made by Pro Oceanus Ltd, Canada (www.pro-oceanus.com). Previous tests with this sensor showed excellent stability in multi-
month vessel-underway missions (Jiang et al., 2014). The instrument float collected data over the top 150 m of ocean depth with an average resolution of 0.3 m from June 2016 to May 2017, while the Pro CV was sampling for 20 minutes at select stop depths (10, 30, 60, 120 m) to allow equilibration with ambient seawater pCO2. These stops resembled bottle stops done from ships with the water rosette to calibrate new sensors. The K1 mooring was also equipped with oxygen sensors to allow for later cross mooring comparisons. The SeaCycler data were corrected for sensor drift using pre- and post-deployment calibration of
the sensors (Atamanchuk et al., 2020). The oxygen data were also corrected for a response time delay using the response time values from Bittig et al. (2014) and the algorithm described in Miloshevich et al. (2004). Overall the accuracy of the oxygen data was $2.89 \pm 4.17 \ \mu M$ based on residuals between the upcast and discrete downcast data. Fully-equilibrated $pCO_2$ data were obtained by averaging the last 30 seconds of the measurements at each stop depth. Accuracy of $pCO_2$ data was determined from the accuracy of the instrument, i.e 0.5 % of the total range.

The glider (Unit 473) was deployed from the Labrador shelf to reach the K1 – SeaCycler site and complete 30 to 100 km long transects between moorings, collecting high-resolution spatial data. The glider was launched near Cartwright, Labrador from a small fishing boat and reached the deep convection zone near K1 and SeaCycler early in October, sampling there until November 22. In total, the glider completed 18 full transects collecting valuable hydrographic and gas data. The modified glider with an extended battery bay carried Sea-Bird glider payload CTD and the Aanderaa Data Instruments (AADI) $CO_2$
optode prototype sensor (model 4797) described in Atamanchuk et al. (2014), and the well established Aanderaa oxygen Optode (Tengberg et al., 2006) model 4831, SN 333. Initial accuracy of the Glider CTD from the manufacturer calibration sheet showed an initial accuracy better than $\pm 0.0005$ S/M, $\pm 0.005$ °C and 0.1% of the total pressure range. Initial accuracy of the calibrated $O_2$ optode from the manufacturer was better than $\pm \ 4 \ \mu M$. Accuracy of the $CO_2$ optode pre-deployment was unknown, but accuracy range in Atamanchuk et al. (2014) is between $\pm \ 2–75 \ \mu atm$. The $CO_2$ optode (SN57) was equipped
with a standard foil to enhance deployment stability. These optode sensors were mounted in the aft cone of the vehicle. Also, a thruster was installed to speed up the crossing of the shelf and to enable staircase profile sampling. The glider sampled the central Labrador Sea deployment location for two months, limiting $CO_2$ optode profiles to the top 200 m to save energy. In December, the glider began its journey back to Newfoundland following the 1500 m isobath inside the Labrador Current and reaching Trinity Bay (see map) on December 31, 2016. The glider was flown along the shelf break to take advantage of the
southward flowing Labrador Current. Before deployment on the glider, the $CO_2$ optode underwent testing at the CERC.OCEAN laboratory at Dalhousie University to determine the calibration model fit for the optode sensor foil.

## 2.2   Trinity Bay Tests

After completion of the VITALS mission, to further test all the characteristics of the new $CO_2$ optode under glider profiling tests, we conducted another study in Trinity Bay, Newfoundland. Trinity Bay is a deep inlet (up to 600 m) and can be reached




easily from various coastal communities from a fishing boat. It is fed primarily by the cold Labrador Current waters and river

runoff from the western side, making its surface waters fresh and deeper portions frigid and highly oxygenated and nutrient-

rich. The pooling of water in the deeper portion and surface freshwater support a stable density stratification (Schillinger et al.,

2000; Tittensor et al., 2002). Especially interesting for our optode tests are the large temperature gradients in the vertical of

over $14°C$ between the surface and 75 m depth. Trinity Bay has a cold water lens $-1°C$ between 70 m to 200 m depth, and

temperatures below $1°C$ from 200 m to the bottom. In Trinity Bay, profiling through this lens leads to absolute temperature

gradients of $10°C$ or more in 200 seconds or less.

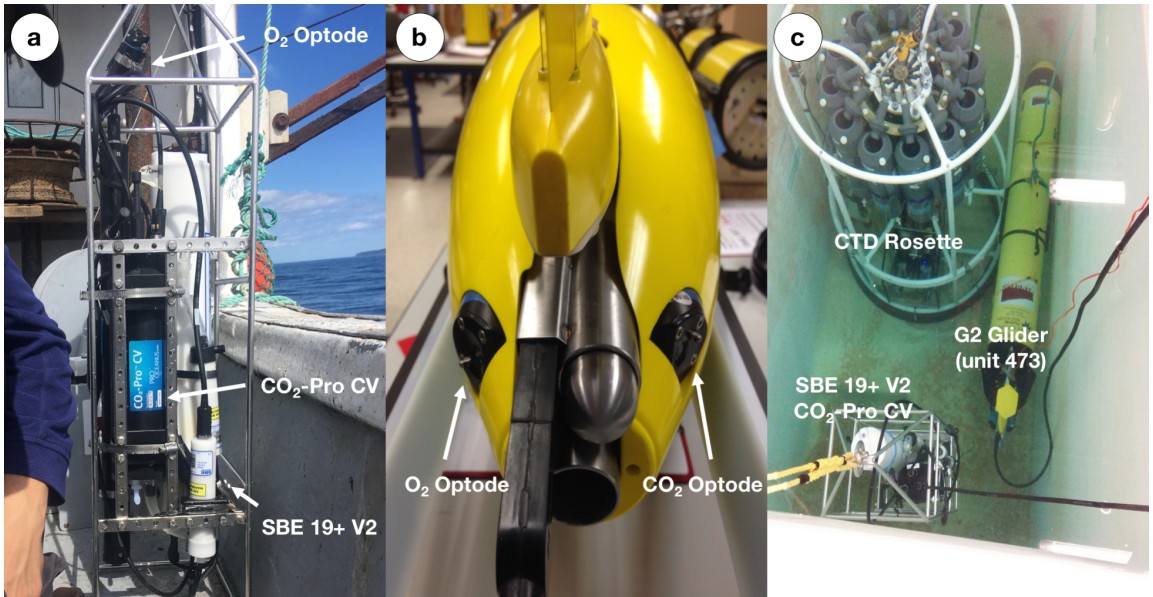

**Figure 2.** (a) Trinity Bay test reference CTD, (b) glider setup and (c) equipment tank testing in progress.

     In Trinity Bay, we repeated the VITALS data comparison experiment on a smaller scale, without the use of a SeaCycler. To

collect in-situ reference samples, we used a winch operated Sea-Bird 19+ V2 CTD mounted on a frame, together with a $O_2$

Optode (Model 4831, SN 333) and a $CO_2$-Pro CV. We repeated step profiles as in VITALS and did extensive calibration of

the sensor prior to and after the deployment. The setups for the external winch-operated CTD and glider are summarized in

Figures 2a and 2b.

     Pre-mission laboratory testing of the sensor and the glider allowed for instrument data quality control in this mission. We

calibrated the $CO_2$ and $O_2$ optode sensors using a double-walled test tank, with simultaneous $O_2$ and $CO_2$ supply for rapid

step changes in these variables. We recorded the optode response in the range of $-1.8°C$ to $20°C$ and $O_2$ concentrations

ranging from 0 to 120% saturation and $CO_2$ concentrations from 100 to 3000 $\mu$atm. Tests were initially done in freshwater and

repeated for 35 ppt NaCl solution. From this tank calibration exercise, new fits for the $O_2$ and $CO_2$ foils were recorded inside

the sensor. Furthermore, tests of all sensors together were done at the special saltwater tank at the Department of Fisheries and

Oceans (DFO) in St. John's, Canada. The tank facility is large enough to allow submerging the glider, CTD-Pro CV setup and





a reference CTD rosette from DFO with an SBE9 CTD, SBE 43 $O_2$ Sensor and Niskin bottles to collect $O_2$ and $pCO_2$ reference
samples for the instruments. Winkler titrations were performed on the Niskin samples to get reference oxygen, DIC and TA
concentrations. DIC and TA were converted to $pCO_2$ using CO2SYS (Lewis et al., 1998). From these measurements and tank
calibration exercises, we computed instrument-specific offsets. For the glider we found CTD residuals were -0.022 $\pm$ 0.0445
$^\circ C$ and -0.081 $\pm$ 0.0153 S/m. For the $O_2$ optode (SN 333) we found offsets of 13.26 $\pm$ 0.493 $\mu M$. A picture of tank testing in
progress is shown in Figure 2c.

## 2.3 Glider Data Processing

We processed glider science data, correcting the CTD-data for temperature-induced sensor lag, applying sequential comparison
between glider profiles Garau et al. (2011). To correct for the phase response lag in the glider oxygen data, we applied the model
published in Bittig et al. (2014) using raw sensor phase angle output. Instead of using the built-in optode thermistor, we used the
lag-corrected CTD temperature readings interpolated to the optode measurements as in Gourcuff (2014). From the corrected
phase readings, we computed the molar oxygen concentrations ($\mu mol \cdot L^{-1}$) using (Uchida et al., 2008), with fit constants from
a prior optode tank calibration. Trinity Bay, tank test and ship-based CTD profiles provided further calibration points at the
start and end of the deployment.

For the $CO_2$ optode, there was some literature available for temperature-dependent response time corrections (e.g. (Bittig
et al., 2014)), but each sensor has its own response time characteristic that has to be determined prior to any field deployments.
Due to the DLR technique in the foil and available field results, the sensor response is larger than the $O_2$ optode, which uses
more straightfoward foil chemistry. To correct for the long response time behaviour, we used a sequential time-lag correction
approach (Miloshevich et al., 2004), recently applied for an equilibrator type NDIR gas instrument (Fiedler et al., 2013). In
Fiedler et al. (2013), the NDIR instrument was mounted on a profiling float, and response times are calculated to be on the
order of 100–300 seconds between surface and depth measurements.

$$c_{i+1}^{cor} = \frac{c_{i+1}^{in\ situ} - [c_i^{in\ situ} \exp(-\Delta t/\tau)]}{1 - exp(-\Delta t/\tau)} \tag{1}$$

Here $c^{in\ situ}$ is the raw and $c^{cor}$ is the corrected sensor output at each time step $i$. The time constant $\tau$ can be computed by fitting
an exponential model to the sensor response $x(t)$ (Equation 2) using fitting constants $a$, $b$ at each time interval $dt$.

$$x(t) = (a - b)\exp(-dt/\tau) + b \tag{2}$$

Atamanchuk et al. (2014) provided a few values for the response time. Also, temperatures were much warmer than found in
the Labrador Sea or Trinity Bay and did not provide response characterization for varying temperature gradients. Fiedler et al.
(2013) used an exponential model (Equation 2) to compare his NDIR sensors response to zero-measurements (ZM). During
ZM's the sensor strips the gas stream of $CO_2$, and the resulting reading should be zero. The time response of the sensor and
resulting reading after ZM were used to gauge the response of the sensor to smaller gas gradients and drift of the gas detector
itself. Because the optode sensor does not have the internal capability for independent referencing of the foil chemistry, we





fitted the equation to the sensor response, while the glider was ascending or descending through the thermocline. Repeating

this procedure for both glider deployments, we computed a temperature and response time-dependent set of values.

Step profiles from Trinity Bay were especially useful for extracting sensor response to a broader set of positive and negative

temperature gradients. Figure 3, shows the least-squares fit for a single temperature gradient and optode response excursion.

To compute the partial pressure of $CO_2$ (p$CO_2$) in micro-atmospheres ($\mu$atm) from the sensors corrected phase readings, we

applied a calibration fit model from previous tank tests as was done in previous deployments of this sensor (Atamanchuk et al.,

2015; Peeters et al., 2016). A testing regime of temperature and molar x$CO_2$ concentrations step changes and sensors phase

response readings were used to compute an 8-degree phase and 3-degree temperature model fit, which we applied to the sensor.

The sensor data and calibration coefficients are available online (von Oppeln-Bronikowski, 2019).

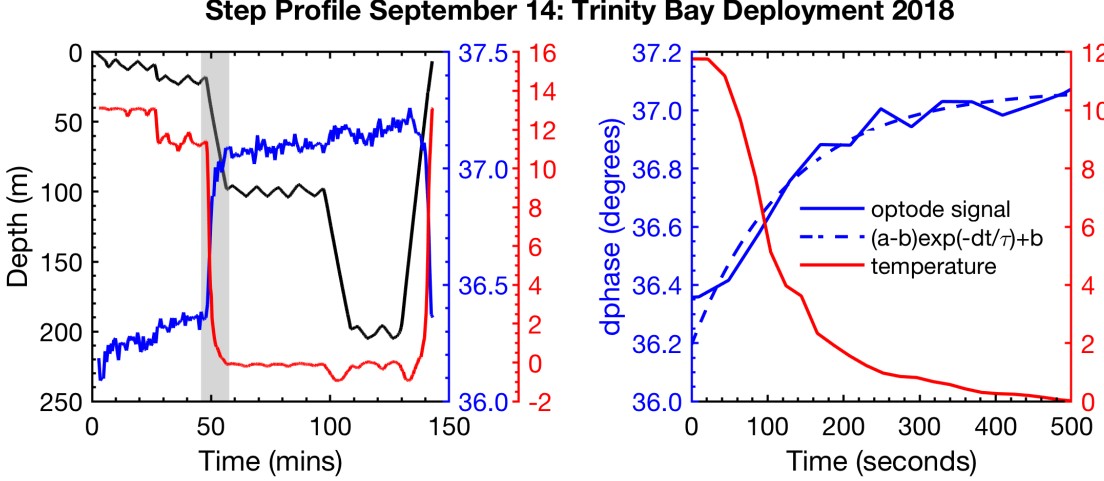

**Figure 3.** Example of a step profile used to quantify response time characteristics. Left panel shows glider step profile (black) overlaid with glider CTD temperature (red) and $CO_2$ optode signal (blue). Grey shaded area highlights an example episode of sensor response, shown in the right panel used to quantify the sensors response time and correct glider profiles. The exponential fit to the $CO_2$ optode response is shown (dashed blue line).

The $CO_2$ optode sensor exhibits noticeable conditioning behaviour (Atamanchuk et al., 2015). For the VITALS deployment,

based on the surface SeaCycler and atmospheric data, we subtracted an offset (1275 $\mu$atm) to correct the the sensor after the

sensor stabilized to ambient conditions. In the VITALS deployment, conditioning took about a month into the deployment,

while in Trinity Bay tests, the sensor response stabilized after roughly a week (994 $\mu$atm). Towards the end of the Trinity Bay

mission, the sensor began to inconsistent drift behavior with depth. Data from the last 1.5 days was excluded from further

analysis. It is not clear what caused this change in the sensors response.

To help with visualization, we bin averaged the data and mapped the data along isopycnals. For some cross-sectional plots,

we also averaged data in depth-space or depth-time sections. To account for the gaps in observations, we preserved gaps larger

than 10 km and more prolonged than 4 days. Smaller gaps were linearly interpolated. A 3D boxcar filter was applied to smooth





5 km in the horizontal, 5 m depth, and 3-day in time, keeping with the observing gaps in the data because the glider occupied a section between K1 and SeaCycler every 2 to 3 days and gaps between profiles were 3 km on average.

To grid the sparse $O_2$ and $pCO_2$ glider observations for spatial-temporal data inter-comparison with SeaCycler, we deviated from linear interpolation. We used an objective interpolation method using a second-degree polynomial fitting distance weighting scheme following Goodin et al. (1979). We gridded the sparse data on a 1-km by 1-day grid and then interpolating the data using an exponential weighting function $\exp\left(R_x^{-2} + R_y^{-2}\right)$ to fill in gaps. We determined influence radii of approximately 5 km for $O_2$ and 20 km for $pCO_2$ measurements and cutoffs at 10 and 40 km respectively, based on the number of glider observations

in the horizontal and along time dimension. We set the cutoff radius at twice the spatial scale. Temporal scales are similar between data sets with an influence radius of 3 days and a cutoff of 6 days.

## 2.4 Shipboard CTD and Pro CV Casts

The Trinity Bay Tests CTD profiles together with $O_2$ optode and data from the Pro CV were processed by checking for outliers in the profiles. Despite the use of a pump, the Pro CV showed long signal equilibration periods ($\tau_{95}$ between 10 to 15 min).

To compute the $CO_2$ levels for each time the CTD was parked at depth, we took the average of the $CO_2$-Pro CV values, once readings stabilized to within $\pm$ 6 ppm or twice the manufacturer's quoted instrument accuracy (0.5% of the total range 0 - 600 ppm). We developed a simple script that identified the first time window when the difference in sensor readings reached $\Delta CO_2 \leq 6$ ppm . Pro CV ZM was subtracted from bottle stops to arrive at a high-quality in-situ referenced data set. We also calculated the standard deviation of each averaged Pro CV measurement and flagged any data points as outliers when the

standard deviation exceeded $\pm$ 6 ppm. Those data points were not included in sensor data comparisons.

## 3 Results and Discussion

### 3.1 Glider-based $CO_2$ Optode Performance

Prior to this mission, the $CO_2$ optode had not been tested on a glider, and little information was available about its response-time characteristics when profiling. We assessed the sensor response time, by fitting the raw sensor signal (dphase) ($\phi_{DLR}$) with

an exponential model, $x(t) = (a - b)\exp\left(-dt/\tau\right) + b$. Here $x(t)$ is the raw sensor signal; $a$ and $b$ are constants, $dt$ is the time interval in seconds, and $\tau$ is the e-folding scale or the response time. Commonly, we define the signal response time, as the time for a signal to reach a specific strength as a percentage of total true signal, we used $\tau_{95}$, that is time to reach 95% of the total signal level. The larger the value, the longer it takes the sensor to equilibrate to ambient conditions. Given the many hundreds of vertical profiles as well as the step profiles taken during the VITALS and Trinity Bay missions, we can do a comparative

analysis of the sensor response time bias against temperature gradient and initial sensor temperature. Figure 4 shows the result of response time fitting against the temperature gradient normalized by the total time of traversing the gradient and the sensor response (e.g. $\tau_{95,\text{normalized}} = \tau_{95}/\Delta T \times 900$). Panel 4a colour indicates the magnitude of root mean square error (RMSE) of least squares data fitting. In panel 4b, color represents the temperature gradient. We multiply normalized values by 900 seconds



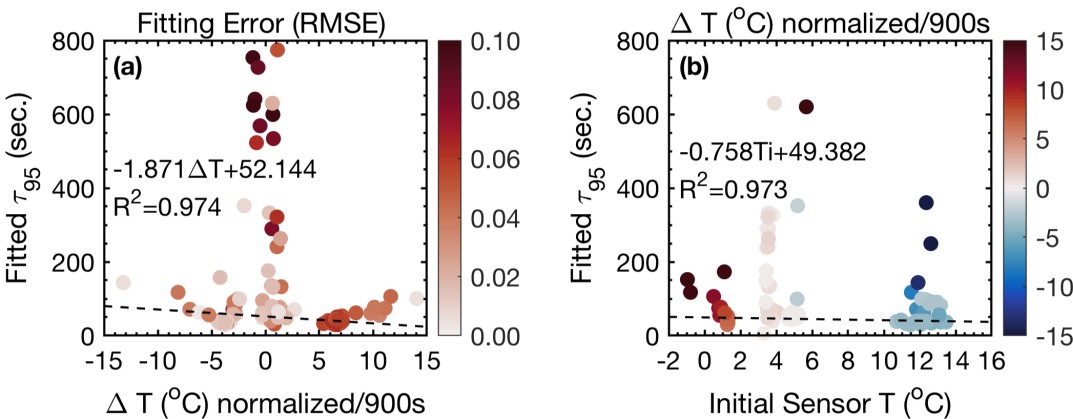

**Figure 4.** (a) $CO_2$ optode response time and temperature gradients colored for RMSE (b) response time and initial sensor temperature colored for $\Delta T$.

or 15 min to arrive at a set of equally referenced temperature gradient and response time values, all corresponding to the same

time interval. We chose this interval based on the response time ($\tau_{95} \approx 15$ min) of the reference sensing system used in the deployments, the Pro CV. We exclude RMSE errors in fits larger than 0.1 (mean is 0.0322 with a standard deviation of 0.0205). There is a strong bias in the response time between warm and cold temperatures, based on the large temperature gradients in Trinity Bay ($< 10°C$). Based on our analysis we find an average sensor response time ($\tau_{95}$ values) of 123.59 sec. with a standard deviation of 181.21 sec. The median response times was 49.20 sec. Minimum values observed were 29.63 sec. and maximum

of 868.02 sec. RMS values significantly increased for fits with response times above 500 seconds. We see large scatter among small gradients from VITALS data, where stratification is less and temperature gradients are small $< 3°C$ compared to Trinity Bay. One would expect shorter response times for smaller temperature gradients. We note here that the VITALS data shown above are mostly derived from average glider profiles (yo's), while Trinity Bay data are mostly step profiles. Also, we see a small trend in initial sensor temperature on response time: that is, an initially colder sensor responds better to warming than a

warm sensor to cooling.

In Figure 4, we showed response time trends of the sensor profiling through weak and strongly temperature stratification and found significant temperature-dependent behaviour. While the temperature-dependence of the sensor foil is non-linear (Atamanchuk et al., 2014), the temperature behaviour of the sensor should show discernible differences when the glider is profiling versus the step mode operation. Superimposing measured raw $CO_2$ optode sensor output with high-quality tempera-

ture, salinity and absolute $CO_2$ measurements from the $CO_2$-Pro CV (Figure 5), we show different response characteristics not captured in Figure 4. For temperature and sensor response time, we expect to see a fairly linear response, as the solubility of $CO_2$ in seawater is reasonably linear (Weiss, 1974) over small temperature ranges ($\Delta T$ less than $10°C$). The unique step profile casts from the Trinity Bay deployment allow an investigation into sensor stability and equilibration, compared to normal glider

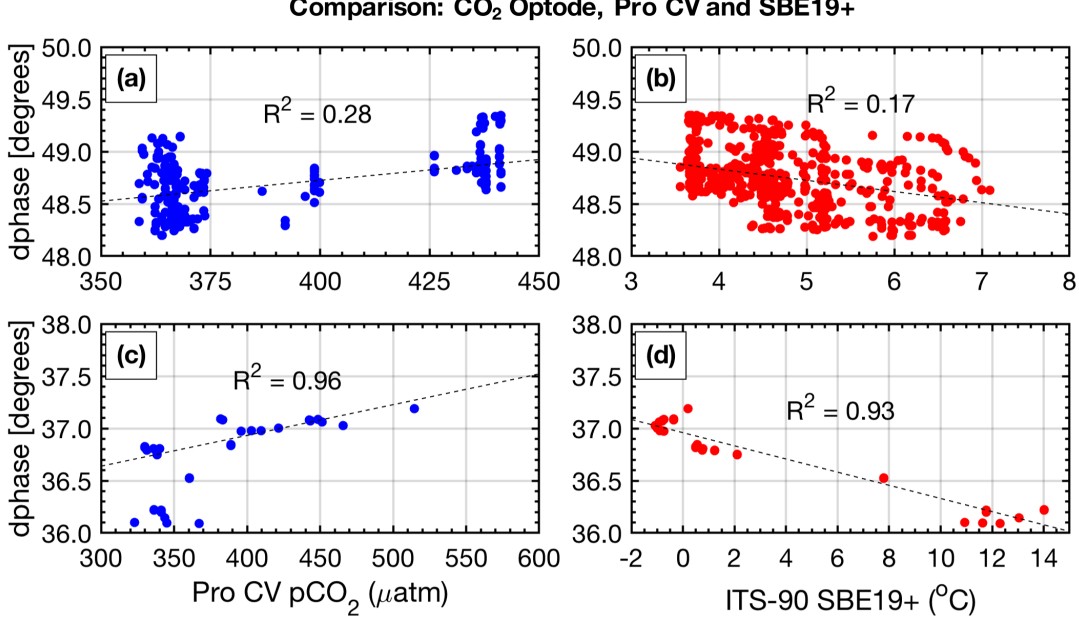

**Figure 5.** $CO_2$ optode signal plotted against VITALS (a) absolute $CO_2$ measured by the Pro CV on SeaCycler and (b) temperature from SBE 19+. Panels (c) and (d) show Trinity Bay step profile data plotted against shipboard measurements. Dashed lines are linear fits.

profiling modes. To match records between observations, we use isopycnal matching, averaging recorded data over bin sizes of

0.01 kg/m$^3$. Figure 5 indeed, shows noticeable differences in the glider $CO_2$ optode data between the two deployments. In the VITALS data, for which we primarily used regular glider profiling, the scatter is much larger among temperature dependence of the response. In contrast, in Trinity Bay data (step profiles), scatter is reduced. Allowing the sensor to fully equilibrate with the ambient conditions, increases the linearity of the sensor response, reproducing the expected linear relationship between $CO_2$ solubility and temperature. For the VITALS data, we see a linear trend in the scatter plot. Still, the spread was not corrected

through our methods giving a broad range of possible $CO_2$ values for a given temperature in the calibration model. More work will be necessary to develop a proper response time model. We also did not consider applying boundary layer and fluid flow model for the optode, such as considered by Bittig et al. (2014) for oxygen optodes. Improvements to the sensor response time as well as more tests are required to evaluate the influence of flow field on the sensor performance.

## 3.2 Comparison: Glider and SeaCycler $O_2$ and $CO_2$ Observations

A novel aspect of the VITALS deployment was the simultaneous measurement of $O_2$ and $CO_2$ from a glider and the SeaCycler profiler, allowing both space and time-varying observations. Given the challenges with validating the glider-based $CO_2$ optode observations, we used the SeaCycler as an in-situ reference for the glider data. For context, the glider and SeaCycler had about two months of overlapping observations. Figure 6 and Figure 7, show the time series data from SeaCycler and monthly

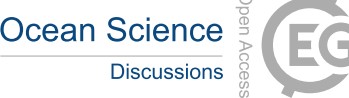

averaged panels from the glider transects. The SeaCycler record is divided into distinct time periods coinciding with changes

to at depth concentration of $O_2$ and $CO_2$. The glider measured both the spatio and temporal evolution of the processes captured

by SeaCycler. Figure 7, shows monthly averaged panels (approximately 10 glider passes distance-averaged per month) of the

glider data. The much lower spatial density of $CO_2$ glider profiles compared $O_2$, means that the $CO_2$ data resolves only spatial

features with scales larger than 20 km, compared to a 5 km resolution for $O_2$. Overall, this region is relatively uniform, with

low spatial gradients. Consistent with the SeaCycler observations, we see a flip between concentrations in $O_2$ and $CO_2$ between

October and November. We also note the different thickness of mixed layer regions across the spatial domain in November.

Smaller pockets of low or high $O_2$ concentrations exist in October, but these trends are weak in an average sense.

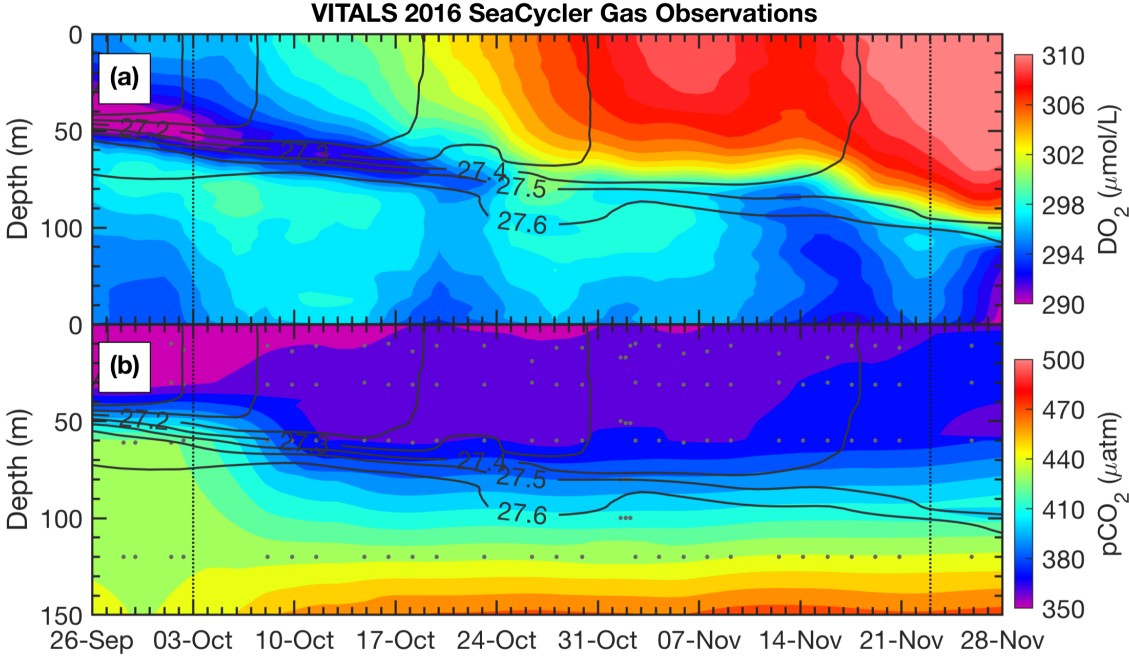

**Figure 6.** SeaCycler time evolution of (a) $O_2$ and (b) $pCO_2$ observations for the joint glider-SeaCycler sampling period with isopycnal anomaly contours overlaid ($0.1\ kg \cdot m^{-3}$ spacing). Small grey dots are the depth and time of discrete $CO_2$ Pro CV measurements by the SeaCycler. Vertical dotted lines indicate the start and end of the joint sampling period.

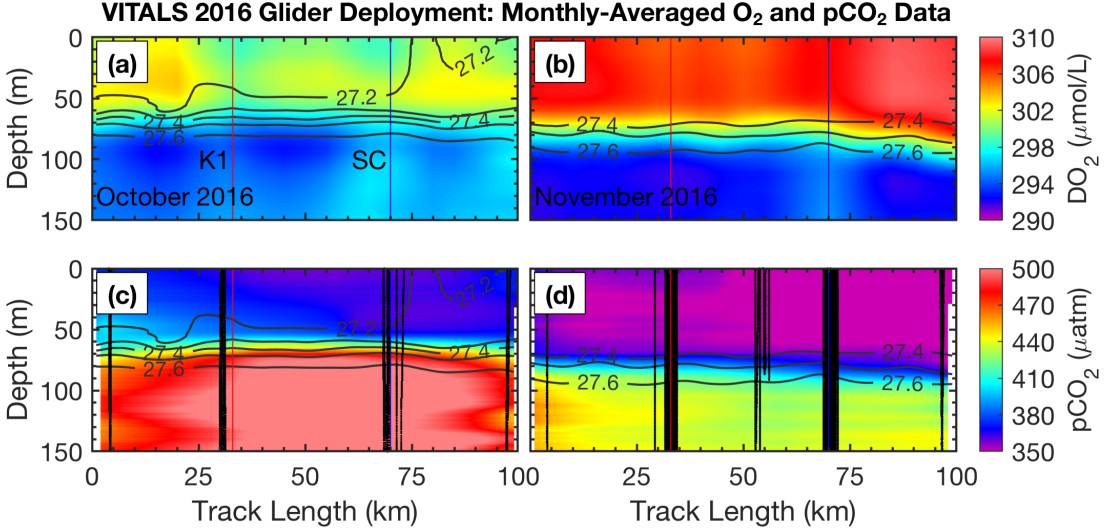

**Figure 7.** Glider monthly averaged spatial section of (a) $O_2$ in October and (b) November and for $pCO_2$ (c) and (d) respectively (along track positions shown in blue inset map in Figure 1). Along-track location of K1 mooring and SeaCycler are indicated with vertical lines as well as individual $CO_2$ optode glider profiles used for plotting.

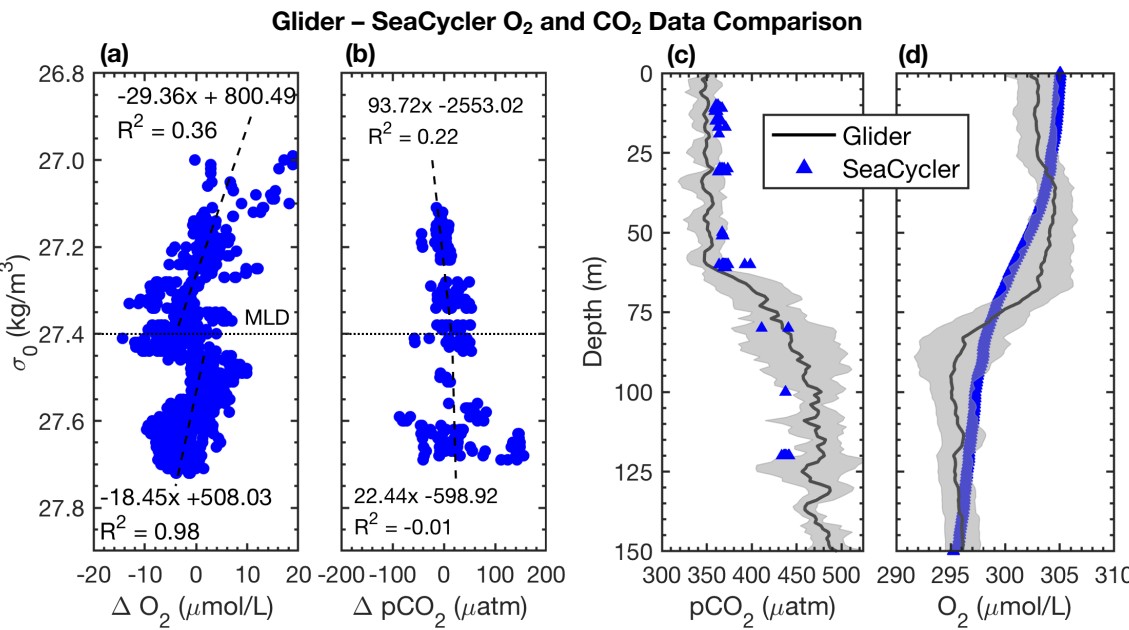

**Figure 8.** Glider-SeaCycler (a) $O_2$ and (b) $pCO_2$ isopycnal-matched residual comparison. Panel (c) and (d) show Glider-SeaCycler corrected depth-averaged $pCO_2$ and $O_2$ values with glider 95% CI shown as grey shading for the period from 3 October to 22 November 2016. Blue triangles are mean of SeaCycler measurements for the glider observing period. Dashed horizontal line in panels (a) and (b) is the average density of the mixed layer and dashed lines are linear fit to the residuals in density space.





To compare the observations between platforms and compute the in-situ reference point for the glider data, we only compared data with similar T–S properties using isopycnal matching. We used the glider and SeaCycler data from the joint observing period (3 October to 22 November), binning data across potential density bins 0.01 kg/m$^3$ to compute the temperature and

salinity residuals from both data sets. If temperature matched to within 0.5°C and salinity to within 0.1 PSU, we allowed these residuals for further comparison of $O_2$ and $pCO_2$ between platform observations. The 95% Confidence Interval (CI) is defined as CI $= \bar{x} \pm 1.96$ STD, where $\bar{x}$ is the average of the variable of interest (e.g. $pCO_2$) and STD is the sample variance. From the matching $O_2$ and $CO_2$ data, we plotted the residual cloud across density and found strong duality in residual trends marked by the 27.56 kg/m$^3$ isopycnal, coinciding with a mixed layer depth (MLD) defined as a change in $\sigma_0 \leq 0.01$ kg/m$^3$.

We used linear, least-squares fits to compute the mean correction of the glider data required to match the SeaCycler (Figure 8) indicating trends above and below the 27.56 kg/m$^3$ isopycnal. Significant scatter ($\pm 50$ $\mu$atm) is observed in $CO_2$ residuals below the mixed layer. Applying the residual fits from the SeaCycler–glider $CO_2$ offsets to the glider data as an in-situ reference (Figure 8c), we see reasonable agreement in the mixed layer. Below the mixed layer, the comparison does not fall within the 95% CI limit. However, we see good agreement and relatively little spread ($\pm 10$ $\mu$mol/L) of $O_2$ data between SeaCycler and

glider sensors leading to a good in-situ reference.

### 3.3 Glider-Observed Spatial and Temporal Variability

Glider-based observations intrinsically link the spatial and time domain, making it hard to differentiate between these two dimensions. In VITALS, we took the approach of doing repeat sections with the glider along the same trajectory to capture both the time and spatial evolution of $O_2$ and $CO_2$ above and below the mixed layer. We applied the residual fits from Figure 8

to the glider data and plotted the Hovmüller diagram of $O_2$ and $pCO_2$ anomalies compared to the SeaCycler data (Figures 9 and 10). Hovmüller diagrams are useful to look at the propagation of processes across the time and space varying field. In this case, we look at how much variability is captured by SeaCycler time-series data along the trajectory sampled by the glider. Because the in-situ comparison between the glider and SeaCycler $CO_2$ data was better at the surface, we only considered the mixed-layer data captured by the glider. We average the glider data top 20 m and grid the observations on a 100 km and 50 days (3 October –

22 November) long track record, subtracting SeaCycler 20 m surface average daily time trend from the glider data. We applied the objective interpolation technique described earlier, interpolating the data using an exponential weighting function to fill in gaps. We could have used linear interpolation for the glider oxygen data but decided to keep mapping methods consistent between $O_2$ and $pCO_2$ data. A drawback of this technique is that it can show artificial variability in the resultant interpolated surface. We applied a low pass filter removing signals shorter than 3-days (time of glider transect) and 4-km (average distance

between dives). We used larger scales of 40 km for the glider $pCO_2$ data. Dots indicate the location of data samples. The legends in the figures, only mask data where no glider data was collected.

We see that there are a few spatial features visible in $O_2$ data. However, the overall spatial structure is not as pronounced as the time variability. Towards the beginning of the record, there is a distinctly more oxygenated zone between K1 mooring and SeaCycler. This could mean that perhaps the low oxygen levels measured by SeaCycler from August to October had more

considerable spatial variability. There are different patterns between moorings. Near SeaCycler, the $O_2$ levels are somewhat




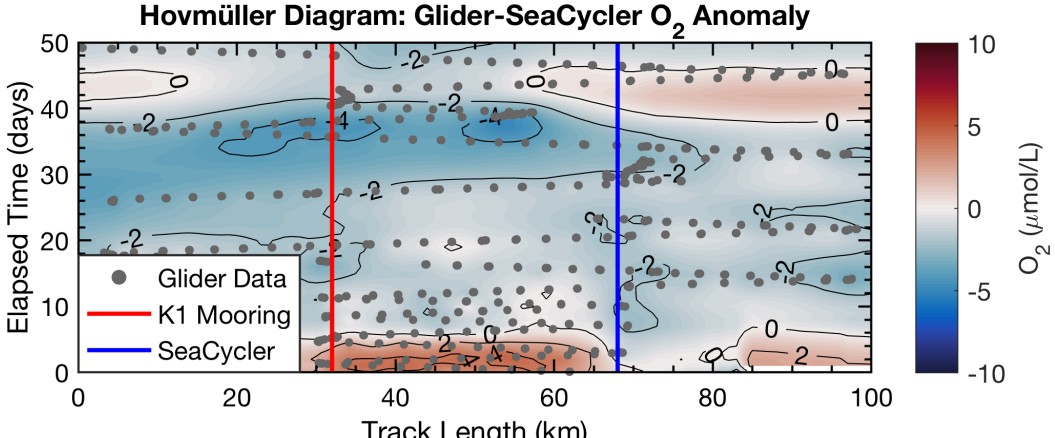

**Figure 9.** Glider Hovmüller diagram, for $O_2$ data (top 20 m) with SeaCycler data removed for period 3 October – 22 November, 2016. Dots indicate the location of data samples. Legends mask area where no glider data was collected.

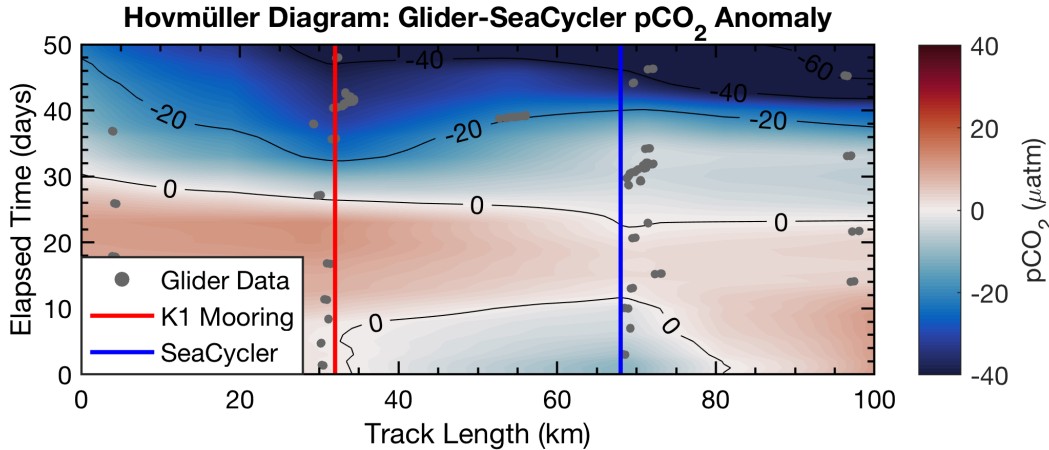

**Figure 10.** Glider Hovmüller diagram, for $CO_2$ data (top 20 m) with SeaCycler data removed for period 3 October – 22 November, 2016. Dots indicate the location of data samples. Legends mask area where no glider data was collected.

higher compared to the K1 mooring, while data near the K1 mooring show lower oxygen levels over time. Towards the second half of the glider record, as storm activity increases in November, the spatial domain becomes smoother. The glider sampled $O_2$ daily and along the entire track length, while the $CO_2$ optode was only sampled at select locations and on average every 2–3 days. The $CO_2$ glider data sampling was too sparse and required too much smoothing to resolve signals smaller than the seasonal cycle. Therefore, the data appears very uniform along the track length. However, this type of direct comparison between platforms will become increasingly important in future glider deployments to achieve long term monitoring capability, recalibrate sensors and quality control mobile platform data.





To separate temporal and spatial variability from another (Figures 9 and 10), we can treat each dimension independently, comparing their autocorrelation scales against each other as a measure of the variability observed. For this analysis, we did not include the glider observations of $CO_2$ due to the sparse sampling across space and time. However, the scales that drive variability in temperature, salinity and $O_2$ data also affect the dynamics of $CO_2$ solubility and the extent and strength of carbon sinks Li et al. (2019); Atamanchuk et al. (2020). We used Chatfield (1998) form of the correlogram or autocorrelation ($r(k)$) as a function of lag $k$.

$$r(k) = \frac{\sum_{t=1}^{N-k}(x_t - \bar{x})(x_{t+k} - \bar{x})}{\sum_{t=1}^{N}(x_t - \bar{x})^2} \tag{3}$$

Here, $x_t$ denotes any quantity of interest (e.g. temperature, salinity or $O_2$) and $\bar{x}$ is the average of $x_t$ along dimension $t$, $k$ can denote either spatial or temporal lags and $N$ is the total number of samples along each dimension. We detrend the gridded space-time glider data to remove non-stationary time and spatial trends following Chatfield and compute the autocorrelation in space and time lags (km and days) for salinity, temperature and $O_2$. We repeat this analysis across the density contours 27.3, 27.7 and 27.75 kg/m$^3$, corresponding closely to surface, intermediate and deepest water regions surveyed by the glider. We include this analysis to the 95% CI bounds defined as $CI = \bar{x} \pm 1.96$ STD, where STD is computed from the range of correlation functions calculated for the whole isopycnal glider space–time data-set.

The autocorrelation function for temperature, salinity and $O_2$ (Figure 11) show different spatial and time scales across all properties between surface and deeper water layers. Temperature, salinity and $O_2$ have similar spatial first zero-crossings of approximately 7–10 km for intermediate and deep waters (27.7–27.75 kg/m$^3$). Oxygen and Temperature also have similar scales (6–7 km) for surface mixed layer waters (27.3 kg/m$^3$). Salinity has first zero crossings of about 10 km scales at the surface mixed layer. CI limits on T and $O_2$ are more similar in intermediate–depth waters and differ at the surface, where T and S seem to be more tightly related, than with oxygen. Across T, S and $O_2$, the CI limit is constrained by 23 km on the upper end and 3 km on the lower end.

Time scales vary more between properties than over spatial scales. Temperature, oxygen and salinity have similar temporal correlation at the surface (11–13 days). On the other hand, oxygen has very different intermediate–depth scales (16 days), compared to T and S (7–11 days) These results suggest that there are different underlying dynamics between the surface and intermediate–deep water layers that drive T, S and $O_2$ time scales as observed in the temporal SeaCycler record (Figure 6. The CI time–scale limits for oxygen are also different compared to T and S in intermediate layers. The temporal scales for oxygen in the intermediate depth layer, fall within the mean of the CI interval (6–29 days), suggesting that the distribution of correlation values is evenly centered around this range of scales. On the other hand T, S scales are closer to the lower limit of the 95% CI bounds (4–24 days).

Overall, spatial scales vary less dramatically between density layers, than temporal scales. The presence of energetic shifting of density layers (every 3 to 5 hours) in the intermediate depth waters would force spatial scales to be small. The glider takes about 3 hours to complete a full dive-climb cycle over a distance of 3–4 km. As the glider begins the next dive-climb cycle, the glider will likely see a shift in the depth of intermediate-depth density layers as it will be between 3 to 5 hours since it first measured the same density layer. A study by Sathiyamoorthy and Moore (2002) explained the observed time scales, looking

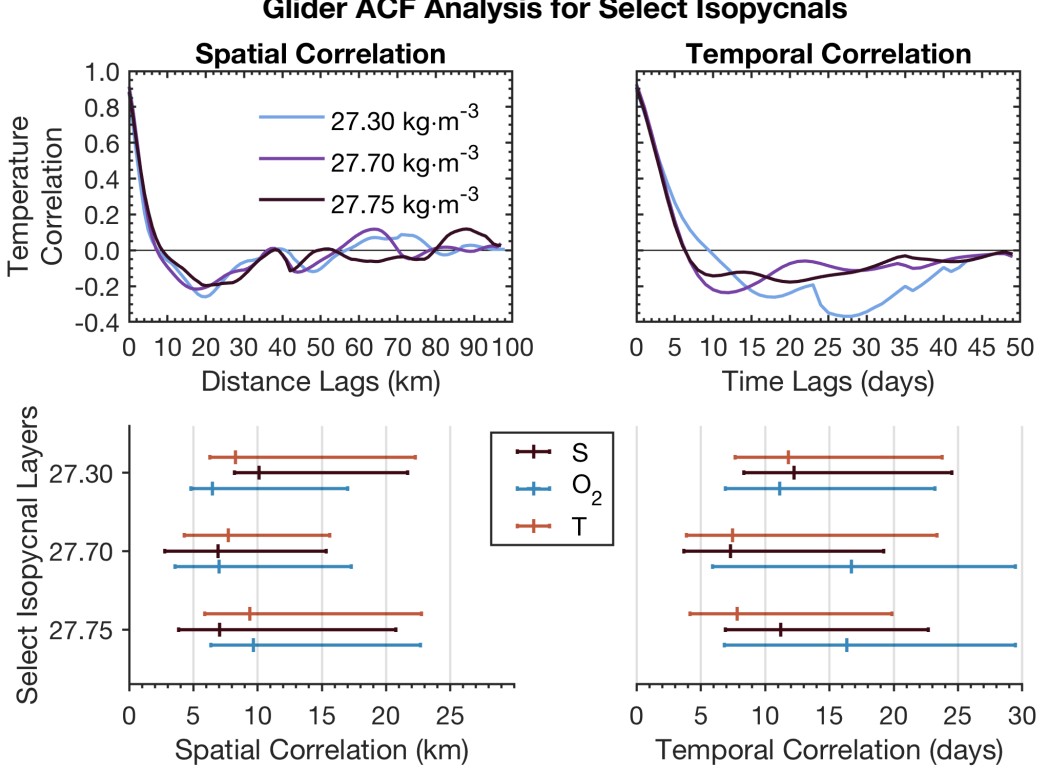

**Figure 11.** Upper panels show example autocorrelation functions for T as a function of distance and time lags. Lower panels summarize zero–crossings for T, S and $O_2$ in space and time lags for shown isopycnals together with 95% CI bounds.

at buoyancy fluxes from OWS Bravo data. Their study found similar time scales of T and S around two weeks at the surface, linking correlation scales in T and S to cyclonic airflow regime changes in the North Atlantic, suggesting storm activity at a period of roughly two weeks in the Labrador Sea in the fall. This result indicates that storms, occurring every few weeks are

primarily responsible for changes in T, S and $O_2$ in the surface layer.

The significant difference in time scales between T–S (7–11 days) and $O_2$ (16 days) across intermediate-deep layers, however, is not intuitive. A possible explanation to this question could be the presence of biological activity that affects $O_2$ at intermediate depth layers, but not T and S. A 2–week period of changes in productivity could be possible, but without further insights from direct observations into the fall and early winter in the Labrador Sea, we can not be sure. The small spatial scales of around 10

km, suggest highly varying changes. Due to temperature dependence, this should also include potential $CO_2$ cycling. We still do not know exactly how much carbon is taken up in the Labrador Sea and understanding the impact of localized changes to solubility pumps is an important step. Small-scale spatial variability of $pCO_2$ for $CO_2$ uptake is important. To distinguish the changes in the strength of $CO_2$ solubility, we need to continue to improve spatial observations of $CO_2$ concentrations in these regions.





We are aware that neither time nor spatial scale results can be interpreted without being mindful of the limitations of the glider platform due to aliasing (Rudnick, 2016). However, compared to contemporary studies in other water regions, our scale results point to much higher variability across all properties, including $CO_2$ along time-space dimensions in the Labrador Sea. For reference, traditional annual ship sampling programs in the Labrador Sea, such as the AR7W section, cover the track of the whole glider mission in a fraction of time, but have spatial gaps on the order of 10s of nautical miles between stations – well

above the average spatial correlation length scales observed by our glider mission. Other platforms such as ARGO floats cover larger areas, but lack the targeted sampling capability offered by glider platforms. Therefore gliders play an important role in constructing an effective observing strategy to resolve the fine-scale processes missed by other platforms.

## 4    Conclusions

In this study, we show data and results of testing a $pCO_2$ optode sensor (Aanderaa model 4979) on a glider. However, improving

the capability of glider-based observations is essential to capture the evolving space–time dynamics of carbon sinks in the ocean. We addressed three questions in this paper: (1) How suitable is the novel $CO_2$ optode for glider-based applications? (2) How can multiple autonomous platforms be used to help improve sensor data? (3) How combined moored and mobile platforms can resolve scales of temporal and spatial variability? We view answering these questions as essential to advance current sensor technology and glider-based $CO_2$ observing capabilities.

Our deployments were the first glider-based tests of the novel $pCO_2$ optode. Several difficulties in using the sensor on a glider were observed such as drift and long response times. We demonstrated the utility of our approach to use staircase missions to improve the quality of sensor data, quantifying more accurately the response time by letting the sensor attain equilibrium with ambient conditions. In both missions, initially the conditioning is followed an steep exponential curve, flattening after some time. In the VITALS mission the sensor showed strong conditioning effects in the first cycle of the deployment and

stabilized after about a month into the deployment. We calculated an initial conditioning offset of 1275 $\mu$atm by comparing the sensor data with atmospheric measurements and SeaCycler. For the Trinity tests the sensor stabilized after about a week, but the sensor showed a non-linear depth-dependent response towards the end of the mission and almost 2 days of data had to be excluded. From the deployments we measured average response times of the sensor in standard glider profiling mode of 123.59 seconds for temperature gradients of 0.5 °C but with a large standard deviation of 181.21 seconds. Using the staircase

referenced optode data, we were able to correct for the response time of the sensor, applying methods similar to those in Fiedler et al. (2013). However, more tests are required to validate our results and characterize the influence of other factors, such as the boundary layer in the sensor's flow field. We identified increased scatter in sensor response times for small temperature gradients (<3 °C). We also detected a small bias in performance towards positive temperature gradients, suggesting the sensor performs better in upcasts than in downcasts. Presently the sensor does not yet have the reliability on its own to measure $pCO_2$

from a glider. The drift and conditioning of the sensor are not well understood and not much prior published test results are available for comparison. It is likely that the sensing foil needs more work to improve stability. The optode has a number of





key strengths, such as its small size, easy integration and low power consumption. If the foil stability and sensitivity could be improved, the sensor could become a desirable candidate for ocean gas measurements similar to the commonly used $O_2$ optode.

In the Labrador Sea mission, we demonstrated how to use the SeaCycler $CO_2$ PRO-CV instrument as an in-situ mid-
deployment reference point to validate the glider $CO_2$ data. Our corrections for the experimental glider $CO_2$ optode, using SeaCycler data yielded a robust surface mixed layer correction of the glider data, but the subsurface data remained noisy. For the more reliable $O_2$ optode, this method worked well, and agreement in data to within $\pm 10$ $\mu$mol/L was achieved.

The unique capability to synchronize and synthesize data from different sensor systems allowed us to investigate the spatial and temporal character of the high-resolution glider observations. The repeat sections of the glider yielded a dynamic picture
across all measured properties (T, S, $O_2$, $CO_2$) in both time and space. On average we observed spatial scales across measured properties of less than 10 km and temporal scales of 15 days or less. We found agreement of our results with previous studies pointing to increased storminess in the fall as an explanation for the roughly 2-week period in time scales. We lacked enough data to also quantify time and spatial scales of $pCO_2$, but given the strong dependence between T and $CO_2$, our results point to the importance of having targeted winter-time glider observations to observe small-scale spatial variability of $CO_2$ cycling.
Overall, our analysis points to much finer scale and localized processes than commonly described in the literature or captured by other observing systems, underlining the importance of repeat glider observations in this region.

These results clearly show that there remain challenges to achieve reliable glider-based $CO_2$ observations. One option is to measure pH rather than $CO_2$. The work by Saba et al. (2018), testing an ISFET pH sensor on a Slocum glider, is indeed very promising. These sensors are already in regular use on BGC Argo floats. Calculation of $pCO_2$ from pH requires knowledge
of at least one other carbonate parameter. On the other hand, pH vs $pCO_2$ relationships measured at fixed platforms like SeaCycler could support this calculation. One limitation of the pH sensor is that one could not use the data to measure air-sea gas exchange, because it is not a direct measurement of $pCO_2$. No matter which sensor one chooses, we believe that in-situ referencing between platforms can add value to existing and future sensors deployments on autonomous platforms such as floats and gliders and add value, at the same time, to the moored measurements.

*Data availability.* VITALS 2016 glider deployment data is available at https://doi.org/10.17882/62358. Processed $CO_2$ optode data from both deployments is available from the authors upon request.

*Author contributions.* NVOB carried out research and initiated the paper, BDY and DA contributed research ideas. All authors contributed to revisions and comments of the paper.

*Competing interests.* None declared.



*Acknowledgements.* We thank Mingxi Zhou, Mark Downey for field work support, Chris L'Esperance with sensor calibration and Fisheries and Oceans Canada for access to their saltwater-tank facility. We thank the National Science and Engineering Research Council (NSERC), Climate Change and Atmospheric Research (CCAR) network for funding this research.



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
