# Peer review of "Glider-Based Observations of CO2 in the Labrador Sea"

_Ocean Science, 2020_

## Referee Comment (RC1) · Anonymous Referee #1 · 4 Sep 2020

General comments: This is a timely article given the necessity for direct CO2 measurements and the rising importance of cross comparison between various platforms. In particular it recognises the importance of subsurface measurements as well as temporal and spatial variability to examine drivers of surface CO2 flux. Direct CO2 measurements made by gliders would be a useful outcome to measure CO2 at depth and in a dynamic environment. The paper deals with comparisons of a novel fast response CO2 sensor with a more established slower response membrane-based sensor. With gliders 100s of profiles are produced for comparison with the step profile of a fully equilibrated sensor. Ultimately, we would hope to increase confidence in the direct fast response CO2 measurements and this paper goes some way towards characterising and validating such measurements. Specific comments: Interesting to learn of a whole

tank validation of gliders and the close attention paid to calibrating sensor foils. I am not convinced that the CO2-CV sensor is ideal for validating data against (and is not necessarily the model referred to in the reference to Jiang et al., 2014). Ideally the tank comparisons would also involve validation with state-of-the-art equilibrator systems. Attention should also be paid to errors in CO2 estimates arising from indirect CO2 estimates (using CO2sys). The paper could also be improved with increased use of tables and attention to detail on the figures (eg: colour legend when required). I have also listed some of the typos (repeated/missing words). After this the paper would be acceptable for publication. Technical: L3. Remove repeat of 'capable' L43. Use carbon sink (not carbon sinks) L44. Plural gliders to remove L53. May not be necessary to spell out CTD but it is an acronym? L56. 'Periods' L59. Inset 'a' (from a..) L87. Fall of 2016 L91. VITALS is an acronym? Fig1 caption requires more detail L102. Clarify that 4797 is the CO2 optode L106. Selected stop depths L108. Validate rather than calibrate L114. Would be good to know precision too? L124. Is CO2 accuracy really 2-75 uatm? It seems a large range (and may depend on the concentration?) L130. Would benefit from putting the dominant current flow onto the map perhaps? L136. Use cold instead of frigid L141. Profile of temperature to capture this? L158. You present T,S,O2 offsets – what about the other variables? A table would help Fig3. Put T and S on the axis (titles and units) L195. Remove duplicate of 'the' L239. Could tabulate some of the response time findings Fig5. Caption could be clearer on what VITALS is so the figure can stand alone L270. To a depth of large change in O2 and CO2? L272. Compared 'to' O2.. Fig6 and Fig7. You switch to DO2 without explanation or reference elsewhere in the text and use just O2 in the caption. Also colour bars/legend required L318. Change 'another' to 'from each other' L327 add year to the Chatfield reference (1998) L337. Its OK to switch to T and O2 but be consistent (in full again on L334)

---

## Referee Comment (RC2) · Anonymous Referee #2 · 12 Sep 2020

Review of Glider-Based Observations of CO2 in the Labrador Sea by: Nicolai von Oppeln-Bronikowski, Brad de Young, Dariia Atamanchuk, and Douglas Wallace

General Comments: The paper is well written and covers a topic of significant interest – testing sensors for different applications and validations against different platforms. In addition, the foray into looking at sub-surface layers in the Labrador Sea is timely and a good demonstration of the usefulness of having pCO2 measurements on gliders. The sensor under test is a CO2 optode sensor previously deployed on static moorings in a variety of locations (by Atamanchuk) and now tested on a glider. The sensor is compared to an infra-red based sensor deployed on an observing platform. The determination of response time of the optode under different glider scenarios (step changes vs profiling) is an interesting and new avenue of investigation and will help

progress the development of the optodes for different platforms. Ultimately the paper shows that the CO2 optode is currently not suitable for glider-based applications, but the use of multiple autonomous platforms can be used to validate sensor performance and provide additional inside into observed processes in the water column. In addition, the temporal and spatial variability observed by the study are evaluated. In general, the author should refer more to the figures in the text, particularly during the discussion sections, as I believe this will help emphasise their conclusions. And use numbers in place of vague terms/adjectives such as somewhat bigger to improve the readability. I think this paper should be published with corrections, which hopefully my queries below can help address. I look forward to seeing this in print.

Specific Comments:

Abstract: I would recommend the addition of some key figures from the text such as optode performance (precision/accuracy), response time, or length of deployment time. These are well utilised in the conclusions so could be used to entice readers within the abstract. The justification of the IR sensors only being used on the SeaCycler is not needed in the abstract, as it is not the focus of the paper and it is not relevant to the abstract to refer to Jiang et al's paper. The few lines discussing the ProCV also adds to the confusion in the last few sentences when referring to "this" and "the" sensor. I also think that the questions posed in the introduction could be summarised more clearly in the abstract.

Introduction I would also suggest the author look at the weather vs climate objectives for sensor performance as defined by GOAON (2nd edition). Line 63 – perhaps the author could make reference to the term "foil". Line 76 – the term "extended" with respect to the trinity bay work implies that it was continuous from the VITALS mission- however further on I understood there was some additional testing between the missions – Am I mistaken? If not perhaps the author could use re-deployed.

Data and Methods Figure 1 – the caption could clarify the importance of red vs. blue

boxes around the profile data. I would also request the profiles on the right hand side have consistent axis (x axis on the top or on the bottom). Are the profiles from the SeaCycler as assuming the blue axis links them to the VITALS work, are there any shipboard CTD's to provide background for the trinity bay? What is the PCO2 sensor CO2 Prototype 4797? Is this the optode? Or is it the SN57? IS this 4797 on the Sea-Cycler – is any data from this presented? Perhaps the authors could clarify why not. With the ProCV can we have some details on how it performs detailed? (e.g. the stability and accuracy calculated from the measurements?). Given that the Jiang study was based aboard a vessel using an underway water supply, are there more references that apply the sensor in situ? If the CO2 optode underwent testing in Dalhousie University before deployment, what was the accuracy and precision also determined prior to deployment? And based on this, the optode should have been partially conditioned under the correct conditions to limit the initial drift of the optode on the glider. Is the SN57 the Aanderaa optode? The Pre-mission testing that was undertaken for the trinity bay work – was this similar undertaken for the VITALS mission? If not this might explain the conditioning timescale difference observed between VITALS and trinity bay. I was also wondering with the inconsistent drift behaviour if there was any other odd responses from other optodes (oxygen) or anything noted on the optode on the post deployment calibration? Perhaps the authors could posit some theories on the behaviour for future investigations. Line 155 – Winkler titrations to not to my knowledge allow you to determine DIC/TA only oxygen. Please can you clarify the instruments used to measure DIC/TA as this may influence the precision/accuracy of these data. In addition, any information on the collection of DIC/TA (e.g. poisoning, storage medium etc). Line 156 – please quote the constants you used for CO2SYS and the errors associated with the calculated pCO2. These will compound any instrument specific offsets.

Glider data processing I like the idea of using the ascent/descent as the ZM from Fiedler. I believe Fiedler also used ZM's to reduced drift of the response – were the authors able to do similar? Was the calibration curved used to calculate pCO2 from the foils from the pre-trinity bay mission testing or from Dariia's paper? Was the same

correction used for both VITALS and trinity bay? Line 179 – remove Also. Line 198 – correct to " the sensor began to display inconsistent behaviour..." (or similar) Line 198 – I would also change the word last to final, as this is clearer as the end of the experiment, rather than a relative statement.

Glider-based CO2 optode performance Response time – I understand the authors are (quite sensibly) looking for relationships between the response time and temperature changes in situ – which is a challenge using only in situ data. The authors undertake a comparative analysis with two parameters, the temperature gradient, and the initial sensor temperature. However, I find the figure 4 (the authors way to display this) confusing. More specifically on Figure 4 – the legend for the colour bar should be next to the bar (ideally rotated) rather than at the top of the figure which implies it is the figure title. I am also not clear on the fitted t95 – is this from the equation listed in the text in line 233? The authors normalise by dividing the temperature change by 900s. I assume deltaT is the minimum and maximum temperatures observed during the 900s intervals? I am not sure what value figure 4a brings as it is not discussed in the text in any detail. The authors then discuss the difference observed between VITALS and trinity bay data – are these both represented in figures 4? If so perhaps a different shape could be used to identify the two cruises while maintaining the figures. Were the response time data for VITALS collected after the sensor had become suitably conditioned to the environment? If not, this could explain the scatter over the smaller temperature gradient. Perhaps the author could expand the sentence that refers to the VITALS glider profiles v.s Trinity Bay step profiles with reference to the response time (or move the sentence closer to the paragraphs below which discuss this in the context of figure 5.) I am also interested in the data points where the response time is above 500s and the variation in RMSE values for these. Perhaps the author could comment on what this means or speculate on why these response times were longer. Is line 244 referring to the relationship shown in figure 4a? Perhaps referring to it in the text and also modifying the spot colour to be the initial sensor T would be helpful here. I am also concerned by the sentence in 246 which states there was a significant temperature dependence on response time. The previous paragraph does not demonstrate this, nor in my opinion figure 4. It would also be good to evaluate this in comparison to Dr Atamanchuk's lab based experiments -they utilised t63 as opposed to t95, but it would be interesting to see the difference between a lab based experiment and an in situ determination – particularly when it appears in figure 4b you have some data collected at 0.5C. Perhaps the authors could clarify the inference from line 252 – while $CO_2$ solubility may have a linear relationship with temperature, I am not convinced that is relevant to the optode response time? Perhaps the author is using this to explain the ProCV response time? I would suggest the author rephase to focus on the sensor response (as a whole) rather than the time taken to respond (as I think this may be their intention). I am not sure if figure 5 a and b are useful plots, as the glider profiling would presumably match less precisely to the CTD-style sea cycler profiles where they remain at the same depth for 20 minutes. The scatter in the data (creating the low r2 on the linear fits) I suspect is also due to the binning implemented to try and maintain a match in data records. I also noted the low R2 values – perhaps the p-values could also be shared to demonstrate that these relationships are indeed significant and provide weight to the "linear trend" statement in line 259. I note the difference in the dphase range between the VITALS and Trinity bay data, yet a not dissimilar $CO_2$ range. The temperature range is significantly wider in trinity bay yet there is no overlap in the dphase values. I was wondering if the author could additionally comment on this (is it a result of the conditioning to local conditions, indicator bleaching?) as it is mentioned only in passing in line 260.

O2-CO2 Observations Please rephrase line 276, as "weak n an average sense" doesn't make sense to me. Figure 7 – the K1 mooring and SeaCycler locations are denoted I think by red and blue lines respectively – these are used within the colour scheme- perhaps white or gray could be considered as alternatives? The O2 data doesn't have the glider profiles used for plotting on? The oxygen data demonstrates the suitability of the multi-platform approach to in situ calibration
Spatial and Temporal Variability Line 310 – please remove the word "somewhat". I would also advise using numbers to make your point clearer. Take care as figure 9/10 the legend appears to obscure data points at the start of a track. Perhaps the legend would be better suited on the right-hand side, or outside the plot. Line 360 -I think it should read "highly variable changes" not "highly varying". The following sentence is also a bit vague – potential CO2 cycling? Perhaps the authors could clarify what they mean by this. Line 363 – I don't think you mean CO2 solubility - do you mean strength of uptake? Or are you referring to the changing T&S increasing or decreasing the solubility?

Conclusion Perhaps the term "staircase missions" could be used in the sections where the authors refer to step profiling to maintain consistency with the conclusion. I would also suggest the author clarify the timescale of the temperature change in line 329 (is it 0.5 degrees over the 123.59 seconds?) I would also suggest that the author summarise some of the extra work, mentioned throughout the rest of the paper as a forward look (e.g. more tests to evaluate the influence of flow field on sensor performance in situ and a response time model?)

---

## Author Response (AR1)

**List of Modifications made to: Glider-Based Observations of CO2 in the Labrador Sea**

Nicolai von Oppeln-Bronikowski et al., 2020

November 12, 2020

**Abstract:**

I would recommend the addition of some key figures from the text…
- Addition of key numbers (sensor accuracy, response time, scales) from results section to engage reader

I also think that the questions posed in the introduction could be summarised more clearly in the abstract
- Summarizing of the main paper goals in more clear way

The justification of the IR sensors only being used on the SeaCycler is not needed
The few lines discussing the Pro CV also adds to the confusion in the last few sentences when referring to "this" and "the" sensor.
- Clarity of text improved by reworking sentences and editorial suggestions as per Reviewer 1 and 2.

**Introduction**

I would also suggest the author look at the weather vs climate objectives for sensor performance as defined by GOAON (2nd edition).
- Added additional references from GOAON report (Newton et al., 2015).

Line 63 – perhaps the author could make reference to the term "foil"
Line 76 – the term "extended" perhaps the author could use re-deployed
- Editorial suggestions as per Reviewer 1 and 2.

**Data and Methods**

Figure 1 – the caption could clarify the importance of red vs. blue boxes around the profile data.
- Improved the Figure 1 layout, labels for clarity and the captions

What is the PCO2 sensorCO2 Prototype 4797? Is this the optode? Or is it the SN57?
- Improved clarity here and elsewhere to distinguish CO2 Pro CV and CO2 Optode.

Winkler titrations to not to my knowledge allow you to determine DIC/TA only oxygen Please can you clarify the instruments used to measure DIC/TA as this may influence the precision/accuracy of these data.  In addition, any information on the collection of DIC/TA (e.g. poisoning, storage medium etc). -      Line 156– please quote the constants you used for CO2SYS

and the errors associated with the calculated pCO2. These will compound any instrument specific offsets.
- Added Appendix section to explain in detail the reference samples taken in the tank test at DFO and uncertainty for the measurements.

L158. You present T, S, O2 offsets – what about the other variables?
- Added table to summarize the tank-test results. Added more sensor accuracy information including Pro CV

L141. Profile of temperature to capture this?
- Figure 2 was improved to include T-S data from Trinity Bay

**Glider Data Processing**

Was the calibration curve used to calculate pCO2 from the foils from the pre-trinity bay mission testing or from Dariia's paper? -        Was the same correction used for both VITALS and trinity bay?
- Improved the clarity on different calibration models applied to the CO2 optode (VITALS vs. Trinity Bay).

Fig3. labels
- Figure 3 labels were improved

**Glider CO2 Optode Performance**

I find the figure 4 (the authors way to display this) confusing.  perhaps the p-values could also be shared to demonstrate Perhaps the author could expand the sentence that refers to the VITALS glider profiles vs. Trinity Bay step profiles
- Improved Figure 4 distinguishing clearly between VITALS and Trinity Tests
- Figure 4b was modified as a box plot for the 2 deployments. The plot complement Table 2.
- Fit was modified to simple linear-least squares with more information on the figure. The fit ignores the VITALS data with large scatter about the origin.

L239.  Could tabulate some of the response time findings
A table was added to summarize more clearly the figure and response time from each deployment

I am also not clear on the fitted t95 – is this from the equation listed in the text in line 233? I would suggest the author rephrase to focus on the sensor response (as a whole) rather than the time taken to respond (as I think this may be their intention).  I am not sure what value figure 4a brings as it is not discussed in the text in any detail.
- We reworded the discussion for clarity and brevity.
- All discussion focussed around improved Figure 4 and Table 2.

I am also concerned by the sentence in 246 which states there was a significant temperature dependence on response time. The previous paragraph does not demonstrate this, nor in my opinion figure 4.
- We fixed the wording with regards to the temperature bias that is shown in Figure 4.

I am not sure if figure 5 a and b are useful plots, as the glider profiling would presumably match less precisely to the CTD-style sea cycler profiles where they remain at the same depth for 20 minutes. The scatter in the data (creating the low r2 on the linear fits) I suspect is also due to the binning implemented to try and maintain a match in data records
- We took out original Figure 5 – agreeing that the figure does not add value in the context of this discussion.

I note the difference in the dphase range between the VITALS and Trinity bay data, yet a not dissimilar CO2 range. The temperature range is significantly wider in trinity bay yet there is no overlap in the dphase values. I was wondering if the author could additionally comment on this (is it a result of the conditioning to local conditions, indicator bleaching?) as it is mentioned only in passing in line 260.
- Mention of the difference added into the text, mentioning possible bleaching.

**O2 and CO2 Observations**

Fig6 and Fig7. You switch to DO2 without explanation or reference elsewhere in the text and use just O2 in the caption. Also colour bars/legend required Figure 7 – the K1 mooring and SeaCycler locations are denoted I think by red and blue lines respectively – these are used within the colour scheme-perhaps white or gray could be considered as alternatives? The O2 data doesn't have the glider profiles used for plotting on?
- Figure labels were modified in (original Figures 6, 7) now Figure 5,6, including color choices and adding location of oxygen glider data on Figure 6.
- Accuracy of glider data from the SeaCycler-glider comparison added
- Table to summarize residuals added

**Glider Observed Spatial and Temporal Variability**

Take care as figure 9/10the legend appears to obscure data points at the start of a track. Perhaps the legend would be better suited on the right-hand side, or outside the plot.
- Figure 8 – Hovmüller Diagram improved, removing legend and using text labels consistent with the rest of the text
- Added uncertainty into the qualitative variability discussion

Please rephrase line 276, as "weak in an average sense" doesn't make sense to me.
- Used numbers in the text to improve clarity of the arguments

Spatial and Temporal Variability Line 310 – please remove the word "somewhat". I would also advise using numbers to make your point clearer. The following sentence is also a bit vague – potential CO2 cycling? Perhaps the authors could clarify what they mean by this.
- Improved legibility of paragraphs when referring to the different scales and interpretation

**Conclusions**

I would also suggest that the author summarise some of the extra work, mentioned throughout the rest of the paper as a forward look(e.g. more tests to evaluate the influence of flow field on sensor performance in situ and a response time model?)
- Reworked second paragraph to summarize extra work done in the study with regards to the goals of the paper

I would also suggest the author clarify the timescale of the temperature change in line 392
- Improved clarity of numbers mentioned in the text.

Following editorial changes were directly implemented in the text
- L3. Remove repeat of 'capable'
- L43. Use carbon sink (not carbon sinks)
- L44. Plural gliders to remove
- L56. 'Periods'
- L59. Insert 'a' (from a..)
- L87. Fall of 2016
- L91. VITALS is an acronym? Fig1 caption requires more detail
- L102. Clarify that 4797 is the CO2 optode
- L106. Selected stop depths
- L108. Validate rather than calibrate
- L114. Would be good to know precision too?
- L136. Use cold instead of frigid
- Line 179 – remove Also.
- L195. Remove duplicate of 'the'
- Line 198– correct to " the sensor began to display inconsistent behaviour..." (or similar)
- Line198 – I would also change the word last to final, as this is clearer as the end of the experiment, rather than a relative statement.
- L270. To a depth of large change in O2 and CO2?
- L318. Change 'another' to 'from each other'
- L327 add year to the Chatfield reference(1998)
- L337. Its OK to switch to T and O2 but be consistent (in full again on L334)
- Line 360 -I think it should read "highly variable changes" not "highly varying".
- Line 363 – I don't think you mean CO2 solubility - do you mean strength of uptake? Or are you referring to the changing T&S increasing or decreasing the solubility?
- Perhaps the term "staircase missions" could be used in the sections where the authors refer to step profiling to maintain consistency with the conclusion.

Author's Response to Reviewer Comments 1

Journal of Ocean Science Interactive Discussion

**Paper Title: *"Glider-Based Observations of CO$_2$ in the Labrador Sea"**

Nicolai von Oppeln-Bronikowski et al.

* Red ink is author response if some other change/ comment was necessary.

Referee #1 Comments

General Comments:

Measurement of the apparent dissociation constants of carbonic acid in seawater at atmospheric pressure

1) **I am not convinced that the CO2-CV sensor is ideal for validating data against (and is not necessarily the model referred to in the reference to Jiang et al., 2014).**
- We will correct the reference and state that it is not the PRO-CV rather the technology that is referenced.

2) **Ideally the tank comparisons would also involve validation with state-of-the-art equilibrator systems. Attention should also be paid to errors in CO2 estimates arising from indirectCO2 estimates (using CO2sys).**
- We will try to consider this in future experiment designs involving this sensor. DIC and TA were estimated in the lab and pCO2 was calculated from CO2Calc (Robbins et al., 2010). TA and DIC are estimated from coulometry (Johnson et al., 1993) and potentiometric titration (Mintrop et al., 2000). In the calculation they used the CO2 equilibrium constants from (Mehrbach et al. 1973 refit. by Dickson and Millero 1987), total boron constant (Lee et al., 2010), and KHSO4 constants (Dickson 1990). I regret the error in the original text which was oversight. The samples were analyzed in the lab. of Fisheries & Oceans Canada and at the moment they are not setup to measure the uncertainty of the pCO2 estimate in CO2calc from DIC and TA. Reported uncertainty in the procedure for DIC and TA were 3 and 4 umol/kg respectively. Unfortuantely CO2calc is not available to me. From repeating the calculations with the same settings mentioned above and using CO2sys, using the uncertainty in TA and DIC, we arrive at an uncertainty of 4.48 uatm for the lab lab-based pCO2 estimates mentioned in the text.

3) **The paper could also be improved with increased use of tables and attention to detail on the figures (eg: colour legend when required).**
- Where appropriate and as pointed out by you in below specific comments, these changes have been implemented. Thank you so much. We modified or slightly adjusted Figures 1-10 with respect to specific comments. We will add a summary table of data mentioned in the text or in the figures near Figure 3 and Figure 4.

Editorial and Other Specific Comments

**L53. May not be necessary to spell out CTD but it is an acronym?**
Noted. In this case given the journal and audience it probably is safe to leave as an acronym.

**L124. Is CO2 accuracy really 2-75uatm? It seems a large range (and may depend on the concentration?)**
Accuracy range is the results so far available in the literature as described in Atamanchuk et al 2014, 2015. The large range, points (in our opinion) to a large range in foil performance under ambient conditions also the range in manufactured foils. Some work better than others…

Temperature and concentration gradients definitely have an impact. Large gradients (see our results) seem to produce more reliably strong signals in the sensor than small gradients. Absolute accuracy is pretty low and foil chemistry was not designed for that. It is not sensitive to absolute concentrations but the change of pH which then induces a fluorescent response of the foil chemistry.

**L130. Would benefit from putting the dominant current flow onto the map perhaps?**
Will try to add arrows. If it is too busy we may omit them as they are not as important to the main story of the paper.

**L141. Profile of temperature to capture this?**
I will add the average T-S structure from Trinity Bay into the paper

**L158. You present T, S, O2 offsets – what about the other variables?**
**A table would help  Fig3. Put T and S on the axis (titles and units)**
Summary tables for CO2 conditioning offsets and improvement to Figure 3 will be implemented in the next revision.

**L239.  Could tabulate some of the response time findings Fig5.  Caption could be clearer on what VITALS is so the figure can stand alone**
Figure 4 and 5 will be modified given feedback from Reviewer 2 and a Table will be used to summarize results from Figure 4.

**L272. Compared 'to' O2..   Fig6 and Fig7.   You switch to DO2 without explanation or reference elsewhere in the text and use just O2 in the caption. Also colour bars/legend required**
Will change DO2 to just O2 to avoid confusion. Will double check if legend placement/colorbar can be improved in the next revision.

All editorial comments/changes below will be addressed in the next revision of the paper.
L3. Remove repeat of 'capable'
L43. Use carbon sink (not carbon sinks)
L44.  Plural gliders to remove
L56. 'Periods'
L59. Insert 'a' (from a..)
L87. Fall of 2016
L91. VITALS is an acronym? Fig1 caption requires more detail
L102. Clarify that 4797 is the CO2 optode
L106.  Selected stop depths
L108.  Validate rather than calibrate
L114. Would be good to know precision too?
L136. Use cold instead of frigid
L195.  Remove duplicate of 'the'
L270. To a depth of large change in O2 and CO2?
L318.  Change 'another' to 'from each other'

L327 add year to the Chatfield reference(1998)
L337. Its OK to switch to T and O2 but be consistent (in full again on L334)

Author's Response to Reviewer Comments 2

Journal of Ocean Science Interactive Discussion

**Paper Title: *"Glider-Based Observations of CO$_2$ in the Labrador Sea"**

Nicolai von Oppeln-Bronikowski et al.

* Red ink is author response if some other change/ comment was necessary.

**Referee #2 Comments**

**Abstract:**

- I would recommend the addition of some key figures from the text such as optode performance (precision/accuracy), response time, or length of deployment time. These are well utilised in the conclusions so could be used to entice readers within the abstract.
  Thank you – results from text have been added to the abstract.

- The justification of the IR sensors only being used on the SeaCycler is not needed in the abstract, as it is not the focus of the paper and it is not relevant to the abstract to refer to Jiang et al. paper.
  Some details will be omitted to add clarity. Also some suggestions from Reviewer 1 will be adapted here.

- The few lines discussing the Pro CV also adds to the confusion in the last few sentences when referring to "this" and "the" sensor. I also think that the questions posed in the introduction could be summarised more clearly in the abstract
  Thank you, abstract has been clarified with regards to the questions from the introduction and to avoid the confusion between mentioned sensors.

**Introduction:**

- I would also suggest the author look at the weather vs climate objectives for sensor performance as defined by GOAON (2nd edition).
  Thank you for the additional source. Additional mention and short sentence to be added into the text.

- Line 63 – perhaps the author could make reference to the term "foil"
  Reference added.

- Line 76 – the term "extended" with respect to the trinity bay work implies that it was continuous from the VITALS mission- however further on I understood there was some additional testing between the missions – Am I mistaken? If not perhaps the author could use re-deployed
  Corrected in text.

**Data and Methods:**

- Figure 1 – the caption could clarify the importance of red vs. blue boxes around the profile data.
  Figure clarified

- I would also request the profiles on the right hand side have consistent axis (x axis on the top or on the bottom).
  Done.

- Are the profiles from the SeaCycler as assuming the blue axis links them to the VITALS work, are there any shipboard CTD's to provide background for the trinity bay?
  Clarified in caption. Profiles are from glider. As per Reviewer 1, Trinity T-S structure added to the text to Figure 2.

- What is the PCO2 sensorCO2 Prototype 4797? Is this the optode? Or is it the SN57? IS this 4797 on the Sea-Cycler – is any data from this presented? Clarify why not
  4797 is the CO2 Optode Prototype sensor which had the manufacturers serial number 57. Clarified text as per Reviewer 1 suggestion

- With the Pro CV can we have some details on how it performs detailed? (e.g. the stability and accuracy calculated from the measurements?).
  The Pro CV had a zero-referencing routine that corrected the drift of the zero point of the sensor (Atamanchuk et al., 2019, supplement section.) Accuracy was given by prior calibration from the manufacturer. Additional references to explain this will be added to the text.

- Given that the Jiang study was based aboard a vessel using an underway water supply, are there more references that apply the sensor in situ?
  We are not aware of many uses of this sensor in-situ. I would gladly include them.

- If the CO2 optode underwent testing in Dalhousie University before deployment, what was the accuracy and precision also determined prior to deployment? And based on this, the optode should have been partially conditioned under the correct conditions to limit the initial drift of the optode on the glider.
  The CO2 optode deployment in VITALS was a first test and we expected stabilization issues. The factory sensor foil calibrations indicated that the sensor met accuracy specifications. In this paper we are reporting on the actual in situ behaviour of the sensor.

- Is the SN57 the Aanderaa optode?
  Yes. Clarified in text.

- The Pre-mission testing that was undertaken for the trinity bay work – was this similar undertaken for the VITALS mission? If not this might explain the conditioning timescale difference observed between VITALS and trinity bay.
  No, there was no prior comparisons done in VITALS that could be used to estimate instrument offsets. This was a motivation for the subsequent tests in Trinity Bay and the lab experiments done with the glider at Fisheries and Oceans. Clarification was added to the text.

- I was also wondering with the inconsistent drift behaviour if there was any other odd responses from other optode (oxygen) or anything noted on the optode on the post deployment calibration? Perhaps the authors could posit some theories on the behaviour for future investigations.
  There was no biofouling or other obstruction found on the sensors during recovery of the gliders. We believe that cold environment, small signal changes (low CO2 gradients) in

the VITALS mission made the sensor response slow and stability low. Also the sensor foils have a large range in performance based on the foil batch. It could be that one foil performs better than another foil calibrated at the same time and same conditions. The oxygen optode meanwhile performed really well and no discernable drift behaviour greater than the accuracy (5 umol/L) was found, although no measurements were collected upon recovery.

- Line 155 – Winkler titrations to not to my knowledge allow you to determine DIC/TA only oxygen
  Fixed in text. TA and DIC are estimated from coulometry (Johnson et al., 1993) and potentiometric titration (Mintrop et al., 2000).

- Please can you clarify the instruments used to measure DIC/TA as this may influence the precision/accuracy of these data.  In addition, any information on the collection of DIC/TA (e.g. poisoning, storage medium etc).
  Instruments used include: VINDTA 3D (Versatile INstrument for the Determination of Total Alkalinity; manufactured by Marianda, Kiel, Germany) DIC analyzer connected to a coulometer (UIC, USA, model 5015O), VINDTA 3S (TA) analyzer using open cell differential potentiometry equipped with a reference (Metrohm, Canada, model 6.0729.100) and pH glass (Thermo-Orion, Canada, model 8101BNWP Ross half-cell) electrode, which were both referenced against a grounded platinum electrode. Samples were collected in the lab in 500 mL BOD bottles and were poisoned 100 uL of saturated Mercuric-Chloride ($HgCl_2$) and allowed to warm in a temperature controlled bath (25C +/- 0.1 C) before analysis.

- Line 156– please quote the constants you used for CO2SYS and the errors associated with the calculated pCO2. These will compound any instrument specific offsets.
  We regret an error in the text. CO2calc (Robbins et al., 1999) and not CO2sys (Lewis and Wallace, 1998) was used in the determining pCO2. In the calculation they used the CO2 equilibrium constants from (Mehrbach et al. 1973 refit. by Dickson and Millero 1987), total boron constant (Lee et al., 2010), and KHSO4 constants (Dickson 1990). The samples were analyzed in the lab. of Fisheries & Oceans Canada and at the moment they are not setup to measure the uncertainty of the pCO2 estimate in CO2calc from DIC and TA. Reported uncertainty in the procedure for DIC and TA were 3 and 4 umol/kg respectively. I do not have access to CO2calc. Using CO2sys with above constants and to repeat the calculations with the uncertainty in TA and DIC, I arrive at an uncertainty of 4.48 uatm for lab experiment pCO2 estimates we reference in the text.

**Glider Data Processing:**

- I like the idea of using the ascent/descent as the ZM from Fiedler. I believe Fiedler also used ZM's to reduced drift of the response – were the authors able to do similar?
  This is an interesting idea and could be tested in a future tank experiment. Using in-situ data it is hard to reach a conclusion on sensor drift and ways to mitigate the effect.

- Was the calibration curve used to calculate pCO2 from the foils from the pre-trinity bay mission testing or from Dariia's paper?
  Separate foil coefficients were used for both missions which were both determined in the CERC laboratory at Dalhousie. Clarified in the text.

- Was the same correction used for both VITALS and trinity bay?
  The conditioning offset from VITALS was estimated by comparison with SeaCycler. The Trinity Bay offsets were not applied as the drift was non linear and a single offset to deal with the conditioning affect was not possible to apply. The same CO2 Optode SN57 was used in both tests. Clarified in text.

**Glider CO2 Optode Performance:**

- I understand the authors are(quite sensibly) looking for relationships between the response time and temperature changes in situ – which is a challenge using only in situ data. The authors undertake a comparative analysis with two parameters, the temperature gradient, and the initial sensor temperature.  However, I find the figure 4 (the authors way to display this) confusing. More specifically on Figure 4 – the legend for the colour bar should be next to the bar (ideally rotated) rather than at the top of the figure which implies it is the figure title.
  Figure 4 will be improved in a variety of ways:
    - We will display a variety of fits (linear least-squares, robust bi-square method as the distribution of the data is not normal but heavily tailed, median and mean responses
    - Colorbars labels will be fixed as per your suggestion

- I am also not clear on the fitted t95 – is this from the equation listed in the text in line 233?
  Yes. Will be clarified.

- The authors normalise by dividing the temperature change by 900s. I assume delta T is the minimum and maximum temperatures observed during the 900sintervals?
  Yes, will be clarified.

- I am not sure what value figure 4a brings as it is not discussed in the text in any detail.
  We understand our initial version was a bit confusing. Figure 4a shows high scatter of response time (or lack of response) at low gradients. Figure 4b shows a slight bias in initial sensor temperature. However, the color coding is perhaps not required.

- The authors then discuss the difference observed between VITALS and trinity bay data – are these both represented in figures 4?
  Yes both are in the figure. Again this will be stated more clearly in the new draft.

- If so perhaps a different shape could be used to identify the two cruises while maintaining the figures.

We will color code the different data sets with colors rather than using colorbars for the RMSE and temperature gradient as no additional information is conveyed.

- Were the response time data for VITALS collected after the sensor had become suitably conditioned to the environment? If not, this could explain the scatter over the smaller temperature gradient.
Yes, this will be added to the text.

- Perhaps the author could expand the sentence that refers to the VITALS glider profiles vs. Trinity Bay step profiles with reference to the response time (or move the sentence closer to the paragraphs below which discuss this in the context of figure 5.)
Sentence will be moved and expanded.

- I am also interested in the data points where the response time is above 500s and the variation in RMSE values for these. Perhaps the author could comment on what this means or speculate on why these response times were longer.
The RMSE is larger for these on average because this data is from VITALS during weak gradients. The good fits come from occasional staircase profiles in that mission that allowed the response to equilibrate. Response times were more scattered as the sensor is not as responsive to weak gradients in temperature (CO2) leading to large tau values for weak gradients.

- Is line 244 referring to the relationship shown in figure 4a? Perhaps referring to it in the text and also modifying the spot colour to be the initial sensor T would be helpful here.
Figure 4b. Will be referenced.

- I am also concerned by the sentence in 246 which states there was a significant temperature dependence on response time. The previous paragraph does not demonstrate this, nor in my opinion figure 4.
Sentence will be rephrased to talk about gradients. Also added fits will help discuss the significance of these results in the context of a variety of fits. A table will be also added to summarize the results.

- It would also be good to evaluate this in comparison to Atamanchuk's lab-based experiments they utilised t63 as opposed to t95, but it would be interesting to see the difference between a lab based experiment and an in situ determination – particularly when it appears in figure 4b you have some data collected at 0.5C.
It is a great idea to have a lab experiment and see my earlier comment on understanding the sensor response in the Glider Processing Section. If we have another opportunity to work with the sensor your point would definitely be included in future comparisons.

- Perhaps the authors could clarify the inference from line 252 – whileCO2 solubility may have a linear relationship with temperature, I am not convinced that is relevant to the optode response time? Perhaps the author is using this to explain the ProCV response

time? I would suggest the author rephrase to focus on the sensor response (as a whole) rather than the time taken to respond (as I think this may be their intention).

Thank you. Good point! Text will be clarified to focus on sensor response to avoid confusion. We mention the linear CO2-T relationship for completeness as another argument to validate the sensor response when true CO2 concentrations are not known.

- I am not sure if figure 5 a and b are useful plots, as the glider profiling would presumably match less precisely to the CTD-style sea cycler profiles where they remain at the same depth for 20 minutes. The scatter in the data (creating the low r2 on the linear fits) I suspect is also due to the binning implemented to try and maintain a match in data records

Good point and I see the argument against having panels a and b. We will play around with the binning. It maybe ok to not include these.

- I also noted the low R2 values – perhaps the p-values could also be shared to demonstrate that these relationships are indeed significant and provide weight to the "linear trend" statement in line 259.

Good point. Will be added and perhaps panels a, b removed from the Figure.

- I note the difference in the dphase range between the VITALS and Trinity bay data, yet a not dissimilar CO2 range. The temperature range is significantly wider in trinity bay yet there is no overlap in the dphase values. I was wondering if the author could additionally comment on this (is it a result of the conditioning to local conditions, indicator bleaching?) as it is mentioned only in passing in line 260.

This is an excellent point and question. We are speculating at this point and not sure for certain what changes the range in the foil angles. This is something we can mention more clearly in the text.

**O2-CO2 Observations:**

- Please rephrase line 276, as "weak in an average sense" doesn't make sense to me.
Numbers added to clarify sentence

- Figure 7 – the K1 mooring and SeaCycler locations are denoted I think by red and blue lines respectively – these are used within the colour scheme-perhaps white or gray could be considered as alternatives? The O2 data doesn't have the glider profiles used for plotting on? The oxygen data demonstrates the suitability of the multi-platform approach to in situ calibration
Figure will be modified for legibility and clarity.

- Spatial and Temporal Variability Line 310 – please remove the word "somewhat". I would also advise using numbers to make your point clearer.
Done and numbers added

- Take care as figure 9/10the legend appears to obscure data points at the start of a track. Perhaps the legend would be better suited on the right-hand side, or outside the plot.
The legend was placed to avoid masking observations by the glider, but we are happy to move the legend outside the figure for clarity

- The following sentence is also a bit vague – potential $CO_2$ cycling? Perhaps the authors could clarify what they mean by this.
Will be clarified. In the text we discuss fine-scale temperature variability, which can contribute to fine-scale variability of $CO_2$ sinks. The complex dynamics that drive ML and below MLD $CO_2$ variability are unfortunately out of scope for this paper.

**Conclusions:**

- I would also suggest the author clarify the timescale of the temperature change in line 392 (is it0.5 degrees over the 123.59 seconds?)
Yes 123.59 seconds is the average time constant we determined from the in-situ data which corresponds to gradients of 0.5 deg C. We will pay attention to the text to clear up any confusing bits.

- I would also suggest that the author summarise some of the extra work, mentioned throughout the rest of the paper as a forward look(e.g. more tests to evaluate the influence of flow field on sensor performance in situ and a response time model?)
Excellent suggestion and the comments provided to the Methods and Sensor Response sections will serve as a good starting point.

Following editorial changes were directly implemented
- Line 179 – remove Also.
- Line 198– correct to " the sensor began to display inconsistent behaviour..." (or similar)
- Line198 – I would also change the word last to final, as this is clearer as the end of the experiment, rather than a relative statement.
- Line 360 -I think it should read "highly variable changes" not "highly varying".
- Line 363 – I don't think you mean $CO_2$ solubility - do you mean strength of uptake? Or are you referring to the changing T&S increasing or decreasing the solubility?
- Perhaps the term "staircase missions" could be used in the sections where the authors refer to step profiling to maintain consistency with the conclusion.

**Glider-Based Observations of $CO_2$ in the Labrador Sea**

Nicolai von Oppeln-Bronikowski[1], Brad deYoung[1], Dariia Atamanchuk[2], and Douglas Wallace[2]

[1]Department of Physics and Physical Oceanography, Memorial University, 283 Prince Phillip Drive, St. John's, NL, A1B3X7, Canada
[2]Department of Oceanography, Dalhousie University, 1355 Oxford Street, Halifax, NS, B3H4R2, Canada

**Correspondence:** Nicolai von Oppeln-Bronikowski (nbronikowski@mun.ca)

**Abstract.** Ocean gliders can provide high spatial and temporal resolution data and target specific ocean regions at a low cost compared to ship-based measurements. An important gap, however, given the need for carbon measurements, is the lack of capable sensors for glider-based $CO_2$ measurements. We need to develop robust methods to evaluate novel $CO_2$ sensors for gliders. Here we present results from testing the performance of a novel $CO_2$ optode sensor (Atamanchuk et al., 2014), deployed on a Slocum glider, in the Labrador Sea and on the Newfoundland Shelf. ~~We demonstrate our concept of validating data from this novel sensor during a long glider deployment using a secondary autonomous observing platform – the SeaCycler. Comparing data between different sensors and observing platforms can improve data quality and identify problems such as sensor drift. SeaCycler carried an extensively tested gas analyzer: the Pro-Oceanus's $CO_2$-Pro CV, as part of its instrument float. The $CO_2$-Pro CV has shown stable performance during lengthy observations e.g. (Jiang et al., 2014), but has a slow response time for continuous profiling, and its power consumption is not affordable for glider operations. This $CO_2$~~ This paper (1) investigates the performance of the $CO_2$ optode on two glider deployments; (2) demonstrates the utility of using the autonomous SeaCycler profiler mooring (Send et al., 2013; Atamanchuk et al., 2020) to improve in-situ sensor data; and (3) presents data from moored and mobile platforms to resolve fine scales of temporal and spatial variability of $O_2$ and $pCO_2$ in the Labrador Sea. The Aanderaa $CO_2$ optode is an early prototype sensor that has not undergone rigorous testing on a glider  but is compact and uses little power.  Our analysis shows that the sensor suffers from instability and slow response times ($\tau_{95}$ >100 s), affected by different behaviour in weak (<0.1 °C) vs. strong (>10 °C) temperature gradients. We compare the glider data with the SeaCycler $O_2$ and $CO_2$ data and estimate the glider data  uncertainty as $\pm$ 6.14 $\mu$M and $\pm$ 44.01 $\mu$atm respectively. From the Labrador Sea data, we point to short time  (<7 days) and distances (<15 km) scales as important drivers of change in this region.

*Copyright statement.* This article is distributed under the Creative Commons Attribution 4.0 License. Unless otherwise stated, associated published material is distributed under the same licence.

[revised manuscript text omitted]